# Column Thresholding for Sparse Spiked Wigner Models: Improved Signal Strength Requirements

**Jian-Feng Cai** [1 2]  **Zhuozhi Xian** [3]  **Jiaxi Ying** [1]

## Abstract

We study the sparse spiked Wigner model, where the goal is to recover an $s$-sparse unit vector $\boldsymbol{u} \in \mathbb{R}^d$ from a noisy observation $\boldsymbol{Y} = \beta \boldsymbol{u}\boldsymbol{u}^\top + \boldsymbol{W}$. While the information-theoretic threshold is $\beta = \widetilde{\Omega}(\sqrt{s})$, existing polynomial-time algorithms require $\beta = \widetilde{\Omega}(s)$, yielding a substantial computational-statistical gap. We propose a column thresholding method that attains the $\widetilde{\Omega}(\sqrt{s})$ scaling for both estimation and support recovery under the non-uniformity condition $\|\boldsymbol{u}\|_\infty = \Omega(1)$. This condition is not merely technical: it explicitly rules out uniform spikes, for which planted-clique-based hardness results apply, and identifies a concrete class of non-uniform spikes where the required signal strength can be reduced. Building on this initializer, we further develop a truncated power method that iteratively refines the estimate with provable linear convergence.

## 1. Introduction

We study the sparse spiked Wigner model (Deshpande & Montanari, 2014), which addresses the problem of recovering a sparse vector $\boldsymbol{u}$ from noisy matrix $\boldsymbol{Y} \in \mathbb{R}^{d \times d}$:

$$\boldsymbol{Y} = \beta \boldsymbol{u}\boldsymbol{u}^\top + \boldsymbol{W}, \tag{1}$$

where $\boldsymbol{u} \in \mathbb{R}^d$ is an unknown $s$-sparse unit vector, $\beta > 0$ denotes the signal strength, and $\boldsymbol{W} \sim \mathrm{GOE}(d)$ is distributed as the Gaussian orthogonal ensemble, i.e., $\boldsymbol{W} = \frac{1}{\sqrt{2}}(\boldsymbol{A} + \boldsymbol{A}^\top)$ with $\boldsymbol{A}$ having i.i.d. $\mathcal{N}(0,1)$ entries. The goal is to estimate $\boldsymbol{u}$ from $\boldsymbol{Y}$ and characterize the regimes

[1]Department of Mathematics, The Hong Kong University of Science and Technology, Hong Kong [2]IAS Center for AI for Scientific Discoveries, The Hong Kong University of Science and Technology, Hong Kong [3]Department of Electronic and Computer Engineering, The Hong Kong University of Science and Technology, Hong Kong. Correspondence to: Jiaxi Ying <jx.ying@connect.ust.hk>.

*Proceedings of the 43$^{rd}$ International Conference on Machine Learning*, Seoul, South Korea. PMLR 306, 2026. Copyright 2026 by the author(s).

of $(\beta, s, d)$ under which nontrivial recovery is information-theoretically and computationally possible. This model captures fundamental inference problems involving pairwise measurements, including Gaussian variants of community detection (Deshpande et al., 2016) and $\mathbb{Z}/2$ synchronization (Javanmard et al., 2016).

**Information-theoretic limits**  The fundamental limits for this model are well-understood. For support recovery and estimation up to constant error, the information-theoretic lower bound on signal strength is $\beta = \widetilde{\Omega}(\sqrt{s})$ (Dia et al., 2016; Lelarge & Miolane, 2017; Banks et al., 2018; Perry et al., 2018; 2020), which is achievable through exhaustive search and other exponential-time procedures, but remains out of reach for polynomial-time algorithms.

**Polynomial-time algorithms**  Existing polynomial-time approaches fall into two main categories, each with fundamental limitations:

*Spectral methods.* The vanilla spectral algorithm computes the leading eigenpair of $\boldsymbol{Y}$, while spectral projection (Brennan et al., 2018) additionally projects this eigenvector onto the set of unit $s$-sparse vectors. Both methods incur $O(d^3)$ computational cost and require signal strength $\beta = \widetilde{\Omega}(\sqrt{d})$ for successful recovery (Baik et al., 2005; Péché, 2006; Féral & Péché, 2007; Paul, 2007; Capitaine et al., 2009; Benaych-Georges & Nadakuditi, 2011; Brennan et al., 2018). This requirement is tight—spectral methods provably fail when $\beta = \widetilde{O}(\sqrt{d})$ (Montanari et al., 2015).

*Thresholding methods.* Diagonal thresholding (Johnstone & Lu, 2009) identifies the support by selecting the $s$ largest diagonal entries of $\boldsymbol{Y}$, then estimates $\boldsymbol{u}$ via the leading eigenvector of the corresponding $s \times s$ submatrix, achieving $O(d \log d + s^3)$ complexity. Covariance thresholding first applies soft (Deshpande & Montanari, 2016) or hard (Krauthgamer et al., 2015) thresholding to all entries of $\boldsymbol{Y}$ before eigendecomposition. While computationally more efficient than spectral methods when $s \ll d$, these approaches all require $\beta = \widetilde{\Omega}(s)$ for recovery (Hopkins et al., 2017; Brennan et al., 2018; Choo & d'Orsi, 2021).

Both the spectral and thresholding methods discussed above fall under a common spectral paradigm. In this framework,

matrix perturbation theory shows that the empirical matrix concentrates around its expectation, which can be viewed as a low-rank "signal" perturbation of a simple baseline (such as the identity); classical eigenvector perturbation bounds then imply that the leading eigenvector aligns with the true spike once the signal is sufficiently strong. This spectral approach is often used as an initialization, followed by a refinement stage, in problems such as phase retrieval (Candes et al., 2015), matrix sensing (Tu et al., 2016), and blind deconvolution (Ma et al., 2018), and such two-stage algorithms are now standard for tackling nonconvex optimization problems (Chi et al., 2019).

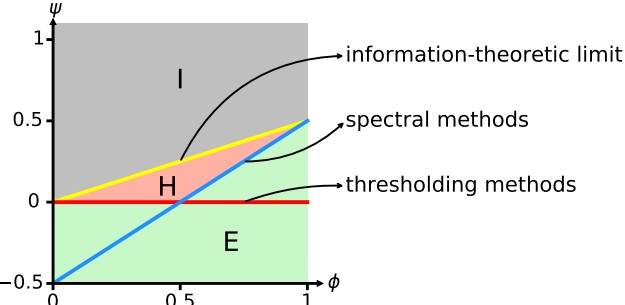

*Figure 1.* Recovery regimes in the sparse spiked Wigner model (Brennan et al., 2018), with $s = \widetilde{\Theta}(d^\phi)$ and $s/\beta = \widetilde{\Theta}(d^\psi)$. Regions I, H, and E correspond to the information-theoretically impossible, computationally hard, and polynomial-time tractable regimes, respectively. The yellow, blue and orange lines show the information-theoretic threshold, the computational boundaries achieved by spectral and thresholding methods, respectively.

**The computational-statistical gap and three regimes** A significant gap exists between the information-theoretic threshold of $\widetilde{\Omega}(\sqrt{s})$ and the $\widetilde{\Omega}(s)$ signal strength required by polynomial-time algorithms. Brennan et al. (2018) characterized this phenomenon through three regimes. Parameterizing $s = \widetilde{\Theta}(d^\phi)$ and $s/\beta = \widetilde{\Theta}(d^\psi)$ with $\phi \in [0, 1]$, these regimes are:

- **Regime I (Impossible):** When $\beta = \widetilde{O}(\sqrt{s})$ (i.e., $\psi > \phi/2$), recovery is information-theoretically impossible for any algorithm (Dia et al., 2016; Lelarge & Miolane, 2017; Banks et al., 2018; Perry et al., 2018; 2020).

- **Regime H (Hard):** When $0 < \psi \leq \phi/2$ and $\psi > \phi - 1/2$, the problem is information-theoretically solvable but conjectured to be computationally intractable in polynomial time under the planted clique hypothesis (Brennan et al., 2018).

- **Regime E (Easy):** When $\beta = \widetilde{\Omega}(s)$ or $\beta = \widetilde{\Omega}(\sqrt{d})$ (i.e., $\psi \leq 0$ or $\psi \leq \phi - 1/2$), polynomial-time algorithms exist. Thresholding methods succeed when $\beta = \widetilde{\Omega}(s)$ ($\psi \leq 0$) (Hopkins et al., 2017; Brennan et al., 2018; Choo & d'Orsi, 2021), while spectral methods succeed when $\beta = \widetilde{\Omega}(\sqrt{d})$ ($\psi \leq \phi - 1/2$) (Benaych-Georges & Nadakuditi, 2011; Brennan et al., 2018).

Assuming the planted clique conjecture holds, the reductions from the planted clique problem show that any polynomial-time algorithms cannot recover a *uniform* spiked vector in the hard regime of Figure 1, where the signal strength lies between $\sqrt{s}$ and $s$. Thus, if one restricts to uniform amplitudes, the statistical–computational gap in this regime is believed to be fundamental and cannot be closed (assuming the conjecture).

By contrast, the computational complexity and the associated phase transitions are much less understood when we consider different *classes of spikes* beyond the uniform case. Our goal is not to claim progress in the classical hard regime for uniform spikes, but rather to clarify what can be achieved once we move beyond the uniform setting. We identify a specific class of spikes, defined by an $\ell_\infty$ lower bound on $\boldsymbol{u}$, in which the uniform vector is ruled out and recovery at the $\sqrt{s}$ signal strength becomes possible.

Under this $\ell_\infty$ condition, we prove that the column thresholding method succeeds at signal strength $\beta = \widetilde{\Omega}(\sqrt{s})$, thereby providing a polynomial-time algorithm that operates in the hard regime for this class of non-uniform spikes, where the planted clique lower bound does not apply. Therefore, our paper does not resolve the planted-clique–hard regime for uniform spikes. Instead, it identifies a different class of spikes, characterized by the $\ell_\infty$ condition, for which recovery at the $\widetilde{\Omega}(\sqrt{s})$ signal strength is provably achievable in polynomial time.

**Our contributions** We propose two algorithms for sparse spike recovery: a column-thresholding method and a truncated power method (TPM) that uses the column-thresholding output as an initialization to refine the estimate. The column-thresholding procedure follows the spectral framework but uses a different statistic: it selects the column with the largest diagonal entry and applies entrywise thresholding, instead of aggregating over all diagonals. This construction yields a stronger separation between in-support and out-of-support indices, which in turn leads to an improved scaling in the signal strength required for successful recovery. Our main contributions are:

- We prove that column thresholding achieves the $\widetilde{\Omega}(\sqrt{s})$ signal-strength scaling for both estimation and support recovery, under the assumption that $\|\boldsymbol{u}\|_\infty = \Omega(1)$. This assumption is not merely technical: it is essential for attaining the $\widetilde{\Omega}(\sqrt{s})$ rate and explicitly rules out the uniform spike case in which planted-clique–based hardness results apply. Our work does not resolve the planted-clique–hard regime for uniform spikes. Rather, by imposing this $\ell_\infty$ condition and analyzing column thresholding, we identify a concrete class of non-uniform spikes, lying outside the reach of existing planted-clique reductions, for which recovery at signal strength $\widetilde{\Omega}(\sqrt{s})$ is provably achievable in polynomial time.

Additionally, the condition $\|\boldsymbol{u}\|_\infty = \Omega(1)$ naturally covers power-law decaying signals (Chen et al., 2015; Jagatap & Hegde, 2019). When this condition fails, our bound degrades to $\widetilde{\Omega}(s)$, matching existing thresholding methods. Conversely, to our knowledge, existing thresholding methods cannot achieve the $\widetilde{\Omega}(\sqrt{s})$ rate even when $\|\boldsymbol{u}\|_\infty = \Omega(1)$ holds.

- We demonstrate that using column thresholding as initialization for truncated power iteration yields a two-stage algorithm with both rigorous guarantees and strong practical performance. While (Yuan & Zhang, 2013) established convergence theory conditional on a correlation condition, they did not provide a concrete initialization procedure. Our work fills this gap by explicitly constructing an initialization that satisfies their theoretical requirements at the optimal signal level, enabling the refinement framework to operate in this previously inaccessible regime.

- Experiments validate our theory and demonstrate strong empirical performance. The column–thresholding method matches the predicted signal–strength scaling, while TPM achieves superior estimation accuracy and exact support recovery compared to baseline methods, all with competitive computational efficiency.

Our results naturally place this work within the study of algorithmic phase transitions and statistical–computational tradeoffs in high-dimensional inference, where Approximate Message Passing (AMP) and the Overlap Gap Property (OGP) offer two important perspectives. Specially, AMP is a powerful framework whose performance can be tracked via state evolution, yielding sharp threshold predictions and, in some cases, Bayes-optimal performance in spiked settings (Montanari & Venkataramanan, 2021). OGP plays a complementary role: it is a structural property—a fragmentation of the near-optimal solution space—widely used to formalize algorithmic hardness in random optimization and inference (Arous et al., 2023; Gamarnik, 2021; Gamarnik & Jagannath, 2021).

It is also natural to extend our methods and analysis to related models. For instance, in the symmetric two-cluster sparse Gaussian mixture model (Pesce et al., 2022; Löffler et al., 2022), the expected sample covariance exhibits the same structural form as in the sparse spiked Wigner model. This analogy allows both diagonal thresholding and our column-thresholding procedure to be used for support estimation of the sparse cluster mean, after which standard eigenvector-based methods can be applied to recover the cluster mean itself.

**Notations:** We use $f(n) = O(g(n))$ when $f(n) \leq c_1 g(n)$, $f(n) = \Omega(g(n))$ when $f(n) \geq c_2 g(n)$, and $f(n) = \Theta(g(n))$ when both hold, for some constants

$c_1, c_2 > 0$. We use $\widetilde{O}, \widetilde{\Omega}, \widetilde{\Theta}$ to denote the logarithm-suppressing variants of $O, \Omega, \Theta$ that hide polylogarithmic factors in $d$. For vector $\boldsymbol{a}$, $a_i$ denotes the $i$-th element, $\|\boldsymbol{a}\|_0$ counts nonzero entries, and $\|\boldsymbol{a}\|_2, \|\boldsymbol{a}\|_\infty$ denote the $\ell_2$ and $\ell_\infty$ norms. Given set $\mathcal{R}$, $\boldsymbol{a}_\mathcal{R}$ zeros out elements indexed by $\mathcal{R}^c$. For matrix $\boldsymbol{A} \in \mathbb{R}^{m \times q}$, $A_{ij}$ is the $(i, j)$-th element. With sets $\mathcal{R}$ and $\mathcal{C}$, $\boldsymbol{A}_{\mathcal{R},\mathcal{C}}$ retains rows in $\mathcal{R}$ and columns in $\mathcal{C}$, zeroing others. Special cases: $\boldsymbol{A}_{:,\mathcal{C}} = \boldsymbol{A}_{\mathcal{R},\mathcal{C}}$ when $|\mathcal{R}| = m$, and $\boldsymbol{A}_\mathcal{R} = \boldsymbol{A}_{\mathcal{R},\mathcal{C}}$ when $\mathcal{C} = \mathcal{R}$.

## 2. Column thresholding

We present a novel column thresholding algorithm for the spiked Wigner model that achieves the information-theoretically optimal signal strength requirement. Our method exploits the key insight that column entries of the observation matrix provide stronger statistical separation than diagonal entries, enabling recovery with signal strength $\beta = \widetilde{\Omega}(\sqrt{s})$ rather than the $\widetilde{\Omega}(s)$ required by existing polynomial-time methods. After developing the algorithm and analyzing its computational complexity, we establish theoretical guarantees for both estimation accuracy and support recovery.

### 2.1. Algorithm

Diagonal thresholding (Johnstone & Lu, 2009) is a well-studied algorithm for the spiked Wigner model that offers low computational cost but requires signal strength $\beta = \widetilde{\Omega}(s)$ for consistent estimation (Hopkins et al., 2017; Choo & d'Orsi, 2021). This requirement significantly exceeds the information-theoretic lower bound of $\beta = \widetilde{\Omega}(\sqrt{s})$ (Dia et al., 2016; Lelarge & Miolane, 2017; Banks et al., 2018; Perry et al., 2018; 2020). We propose a novel thresholding algorithm that reduces the required signal strength.

To understand the limitations of diagonal thresholding, we analyze its signal strength requirements. The algorithm estimates the support of $\boldsymbol{u}$ by selecting indices corresponding to the $s$ largest diagonal entries of $\boldsymbol{Y}$, then computes the leading eigenvector of the resulting submatrix. This approach exploits the expected diagonal structure:

$$\mathbb{E}[\boldsymbol{Y}]_{ii} = \begin{cases} \beta |u_i|^2, & i \in \mathcal{T}, \\ 0, & i \in \mathcal{T}^c, \end{cases} \tag{2}$$

where $\mathcal{T}$ is the support of $\boldsymbol{u}$. The statistical gap between in-support and out-of-support entries is

$$g_{\text{diag}} := \min_{i \in \mathcal{T}} \mathbb{E}[\boldsymbol{Y}]_{ii} - \max_{i \in \mathcal{T}^c} \mathbb{E}[\boldsymbol{Y}]_{ii} = \beta \cdot \min_{i \in \mathcal{T}} |u_i|^2. \tag{3}$$

The following proposition shows when diagonal thresholding successfully identifies the support:

**Proposition 2.1.** *If $|W_{jj}| \leq \frac{1}{2} g_{\text{diag}}$ holds for all $j \in [d]$, then $Y_{ii} > Y_{i'i'}$ for all $i \in \mathcal{T}$ and $i' \in \mathcal{T}^c$.*

The proof of Proposition 2.1 is provided in Appendix A.3. Since $W_{jj} \sim \mathcal{N}(0, 2)$ independently, the condition holds with high probability when $g_{\text{diag}}$ is sufficiently large. In that case, the diagonal entries $Y_{ii}$ for $i \in \mathcal{T}$ exceed those for $i \in \mathcal{T}^c$, so diagonal thresholding recovers the support of $\boldsymbol{u}$ by selecting the largest $s$ diagonal entries of $\boldsymbol{Y}$. Thus, support recovery reduces to ensuring a sufficiently large separation between the two sets of diagonal entries. The success probability is governed by the gap $g_{\text{diag}}$: a larger gap $g_{\text{diag}}$ yields a higher probability of correctly estimating the support. Alternatively, when $g_{\text{diag}}/\beta$ is large, achieving any target gap sufficient for recovery requires less $\beta$.

Our key insight is to leverage column entries instead of diagonal entries to achieve better separation between in-support and out-of-support indices. Our approach is based on the expected $l$-th column when $l \in \mathcal{T}$:

$$\mathbb{E}\big[\boldsymbol{Y}\big]_{il} = \begin{cases} \beta u_i u_l, & i \in \mathcal{T}, \\ 0, & i \in \mathcal{T}^c. \end{cases} \tag{4}$$

The resultant gap becomes:

$$g_{\text{col}} := \min_{i \in \mathcal{T}} \big|\mathbb{E}\big[\boldsymbol{Y}\big]_{il}\big| - \max_{i \in \mathcal{T}^c} \big|\mathbb{E}\big[\boldsymbol{Y}\big]_{il}\big| = \beta |u_l| \min_{i \in \mathcal{T}} |u_i|.$$

Crucially, $g_{\text{col}} = \beta |u_l| \min_{i \in \mathcal{T}} |u_i| \geq \beta \min_{i \in \mathcal{T}} |u_i|^2 = g_{\text{diag}}$ whenever $l \in \mathcal{T}$, providing enhanced separation that enables recovery with weaker signal strength requirement. To maximize the gap $g_{\text{col}}$, $l$ should ideally be the index of the largest absolute element of $\boldsymbol{u}$, which aligns with the index of the largest diagonal entry of $\mathbb{E}\big[\boldsymbol{Y}\big]$, as shown in (2). However, since we only have the noisy matrix $\boldsymbol{Y}$, we choose $l$ as the index of the largest diagonal entry of $\boldsymbol{Y}$, denoted by $i_0$.

---

**Algorithm 1** Column Thresholding

1: **Input:** Matrix $\boldsymbol{Y} \in \mathbb{R}^{d \times d}$, sparsity level $s$
2: $i_0 \leftarrow \arg\max_{i \in [d]} Y_{ii}$
3: $\widehat{\mathcal{T}} \leftarrow$ indices of $s$ largest entries in $|\boldsymbol{Y}_{:,i_0}|$
4: $\boldsymbol{Y}_{\widehat{\mathcal{T}}} \leftarrow$ submatrix of $\boldsymbol{Y}$ with rows and columns in $\widehat{\mathcal{T}}$
5: $\boldsymbol{v} \leftarrow$ leading eigenvector of $\boldsymbol{Y}_{\widehat{\mathcal{T}}}$ with $\|\boldsymbol{v}\|_2 = 1$
6: Initialize $\hat{\boldsymbol{u}} \leftarrow \boldsymbol{0} \in \mathbb{R}^d$ and set $\hat{\boldsymbol{u}}_{\widehat{\mathcal{T}}} \leftarrow \boldsymbol{v}$
7: **Output:** Estimated sparse unit vector $\hat{\boldsymbol{u}} \in \mathbb{R}^d$

---

**Algorithm 2** Column Thresholding (Normalization Variant)

1: **Input:** Matrix $\boldsymbol{Y} \in \mathbb{R}^{d \times d}$, sparsity level $s$
2: $i_0 \leftarrow \arg\max_{i \in [d]} Y_{ii}$
3: $\widehat{\mathcal{T}} \leftarrow$ indices of $s$ largest entries in $|\boldsymbol{Y}_{:,i_0}|$
4: Set $\hat{\boldsymbol{u}}_{\text{nv}} \leftarrow \boldsymbol{Y}_{\widehat{\mathcal{T}}, i_0} / \|\boldsymbol{Y}_{\widehat{\mathcal{T}}, i_0}\|_2$
5: **Output:** Estimated sparse unit vector $\hat{\boldsymbol{u}}_{\text{nv}} \in \mathbb{R}^d$

---

Algorithm 1 implements our column thresholding approach in two steps: (1) estimate the support $\widehat{\mathcal{T}}$ using the $s$ largest entries of the $i_0$-th column, where $i_0 = \arg\max_i Y_{ii}$; (2) reconstruct the spike vector using the leading eigenvector of the $s \times s$ submatrix $\boldsymbol{Y}_{\widehat{\mathcal{T}}}$, formed by restricting to rows and columns indexed by $\widehat{\mathcal{T}}$. For computational efficiency, Algorithm 2 presents a variant that directly normalizes the selected column entries.

The enhanced statistical gap in our column-based approach is the key to achieving the optimal signal strength requirement of $\beta = \widetilde{\Omega}(\sqrt{s})$. As detailed in Section 3.2, this improvement stems from leveraging correlations among entries in the selected column, which provide stronger signal concentration than the independent diagonal entries used in diagonal thresholding. This idea is in line with prior works that use structure-aware initialization, thresholding/truncation, or tailored refinement to obtain sample-efficient guarantees for sparse phase retrieval (Jagatap & Hegde, 2019; Wu & Rebeschini, 2021; Cai et al., 2023) and the spiked covariance model (Cai et al., 2025). Compared with these works, our contribution is to adapt this principle to the spiked Wigner model and establish an improved recovery guarantee.

Algorithm 2 presents a computationally efficient variant that applies the same thresholding strategy for support estimation but directly normalizes the $i_0$-th column rather than computing an eigenvalue decomposition. This variant trades modest estimation accuracy for reduced computational cost while preserving the optimal signal strength requirement of $\beta = \widetilde{\Omega}(\sqrt{s})$ and maintaining the same theoretical guarantees for support recovery. The variant is practical when computational resources are limited or rapid support identification is prioritized over exact reconstruction.

### 2.2. Computational Complexity

Algorithm 1 requires three procedures: finding the largest diagonal entry ($O(d)$), selecting the $s$ largest column entries ($O(d \log d)$ via sorting or $O(d + s \log s)$ using partial sorting), and computing the leading eigenvector of an $s \times s$ matrix ($O(s^3)$). The total complexity is $O(d \log d + s^3)$, which reduces to $O(d \log d)$ when $s = O((d \log d)^{1/3})$—a regime covering many practical sparse recovery scenarios. In comparison, the normalization variant (Algorithm 2) eliminates the eigendecomposition step, achieving $O(d \log d)$ complexity.

For comparison, diagonal thresholding (Johnstone & Lu, 2009) selects the $s$ largest diagonal entries and computes the top eigenvector of the resulting $s \times s$ submatrix, for $O(d \log d + s^3)$ time. In contrast, spectral methods (vanilla spectral and spectral projection (Brennan et al., 2018)) compute the leading eigenvector of the full $d \times d$ matrix $\boldsymbol{Y}$, requiring $O(d^3)$ operations.

Column thresholding improves the signal-strength scaling for consistent recovery. Whereas diagonal thresholding and spectral methods both require suboptimal signal strength $\beta = \widetilde{\Omega}(s)$ for consistent recovery, our column thresholding succeeds at $\beta = \widetilde{\Omega}(\sqrt{s})$ under mild conditions. At the same time, we maintain the computational efficiency of diagonal thresholding while still offering substantial speedup over spectral methods.

### 2.3. Theoretical analysis

We establish theoretical guarantees showing that column thresholding succeeds at signal strength $\beta = \widetilde{\Omega}(\sqrt{s})$ under mild conditions. We analyze estimation accuracy and support recovery separately.

#### 2.3.1. ESTIMATION ERROR

We analyze estimation accuracy using the following distance metric accounting for sign ambiguity:

$$\text{dist}(\boldsymbol{u}, \hat{\boldsymbol{u}}) := \min\left\{ \|\boldsymbol{u} - \hat{\boldsymbol{u}}\|_2, \|\boldsymbol{u} + \hat{\boldsymbol{u}}\|_2 \right\}. \qquad (5)$$

This metric is standard in PCA and phase retrieval where the sign of the recovered vector is inherently ambiguous.

**Theorem 2.2.** *Let $\boldsymbol{u} \in \mathbb{R}^d$ be an $s$-sparse unit vector and $\boldsymbol{Y} = \beta \boldsymbol{u}\boldsymbol{u}^\top + \boldsymbol{W}$, where $\beta > 0$ and $\boldsymbol{W} \in \mathbb{R}^{d \times d}$ is distributed as $\text{GOE}(d)$. For any target accuracy $\zeta \in (0, 1]$, if the signal strength satisfies*

$$\beta \geq C_1 \zeta^{-1} \|\boldsymbol{u}\|_\infty^{-1} \sqrt{s \log d}$$

*for some universal constant $C_1 > 0$, then with probability at least $1 - 1.6d^{-1}$, the output $\hat{\boldsymbol{u}}$ of Algorithm 1 satisfies $\text{dist}(\boldsymbol{u}, \hat{\boldsymbol{u}}) \leq \zeta$.*

This theorem, proved in Appendix A.4, shows that column thresholding achieves signal strength scaling of $\Omega(\sqrt{s \log d})$ for constant estimation error when $\|\boldsymbol{u}\|_\infty = \Omega(1)$, matching the information-theoretic limit of $\widetilde{\Omega}(\sqrt{s})$. This improves the $\widetilde{\Omega}(s)$ signal strength requirement of existing polynomial-time algorithms, including diagonal thresholding and spectral methods.

Attaining the optimal signal strength $\widetilde{\Omega}(\sqrt{s})$ in polynomial time requires additional assumptions. Indeed, computational hardness results based on the planted clique conjecture indicate that no polynomial-time algorithm can achieve the optimal signal strength without additional structural assumptions (Brennan et al., 2018). This infinity norm condition is mild and naturally satisfied in many applications. For instance, when the nonzero entries of $\boldsymbol{u}$ follow a power-law decay—a common model in compressive sensing (Donoho, 2006; Candès et al., 2006)—the infinity norm requirement is automatically satisfied. Similar phenomena arise in sparse phase retrieval, where power-law signals enable optimal recovery (Jagatap & Hegde, 2019).

The role of non-uniformity in our result is explicit and continuous. Specially, Theorem 2.2 gives a guarantee at signal strength $\widetilde{\Omega}(\sqrt{s}\|\boldsymbol{u}\|_\infty^{-1})$ throughout the admissible range of $\|\boldsymbol{u}\|_\infty$. In the uniform case $\|\boldsymbol{u}\|_\infty = s^{-1/2}$, the requirement becomes $\widetilde{\Omega}(s)$, which matches the known easy regime in Figure 1 and is therefore fully consistent with planted-clique-based hardness. By contrast, when $\|\boldsymbol{u}\|_\infty = \Omega(1)$, our method succeeds at $\widetilde{\Omega}(\sqrt{s})$, matching the information-theoretic scaling known for the uniform case. Intermediate regimes interpolate smoothly, e.g. $\widetilde{\Omega}(s^{1/2+\alpha})$ when $\|\boldsymbol{u}\|_\infty \asymp s^{-\alpha}$ for some $\alpha \in (0, 1/2)$.

**Theorem 2.3.** *Under the same model assumptions as Theorem 2.2, if the signal strength satisfies*

$$\beta \geq C_2 \zeta^{-2} \|\boldsymbol{u}\|_\infty^{-1} \sqrt{s \log d}$$

*for some universal constant $C_2 > 0$, then with probability at least $1 - 1.4d^{-1}$, the output $\hat{\boldsymbol{u}}_{\text{nv}}$ of Algorithm 2 satisfies $\text{dist}(\boldsymbol{u}, \hat{\boldsymbol{u}}_{\text{nv}}) \leq \zeta$.*

The proof of Theorem 2.3 is provided in Appendix A.5. While the normalization variant (Algorithm 2) requires a stronger dependence of $\beta$ on the accuracy parameter $\zeta$ (quadratic rather than linear), it maintains the optimal scaling with respect to $s$ and $d$. This highlights a key insight: the statistical efficiency is determined by the column thresholding step for support estimation, not by the reconstruction method (eigendecomposition versus direct normalization). The reconstruction only affects the constant factors in the accuracy guarantee, confirming that our column-based support estimation is the fundamental innovation enabling optimal signal strength requirements.

#### 2.3.2. SUPPORT RECOVERY

Beyond estimation accuracy, exact support recovery is crucial for interpretability and downstream tasks that require identifying the active variables. Let $\boldsymbol{u} \in \mathbb{R}^d$ be an unknown $s$-sparse unit vector with support

$$\mathcal{T} := \{\, i \in [d] : u_i \neq 0 \,\}, \qquad |\mathcal{T}| = s.$$

We establish conditions under which our proposed algorithms output an estimated support $\widehat{\mathcal{T}}$ satisfying $\widehat{\mathcal{T}} = \mathcal{T}$ with high probability.

**Theorem 2.4.** *Let $\boldsymbol{u} \in \mathbb{R}^d$ be an $s$-sparse unit vector satisfying $|u_i| \geq \theta/\sqrt{s}$ for all $i \in \mathcal{T}$ and some constant $\theta > 0$. Under the spiked Wigner model with signal strength*

$$\beta \geq C_3 \theta^{-1} \|\boldsymbol{u}\|_\infty^{-1} \sqrt{s \log d}$$

*for some universal constant $C_3 > 0$, both Algorithms 1 and 2 recover the support exactly (i.e., $\widehat{\mathcal{T}} = \mathcal{T}$) with probability at least $1 - 1.3d^{-1}$.*

The proof of Theorem 2.4 appears in Appendix A.6. Our support recovery guarantee attains the signal strength scaling of $\Omega(\sqrt{s \log d})$, matching the information-theoretic limits $\widetilde{\Omega}(\sqrt{s})$. The minimum magnitude assumption $|u_i| \geq \theta/\sqrt{s}$ ensures that all nonzero entries are sufficiently strong to be distinguished from noise, as detailed in Appendix A.6. This condition is standard in the sparse recovery literature and naturally holds for many structured signals. Since Algorithms 1 and 2 differ only in their estimation procedures while using identical support recovery methods, they achieve the same support recovery guarantees.

## 3. Truncated power method

Column thresholding meets the optimal signal-strength requirement but can benefit from iterative refinement to improve accuracy. Its one-shot estimate, though supported by strong theory, may not attain the smallest achievable error. In this section, we show how the truncated power method iteratively refines the initial estimate, yielding improved estimation accuracy.

In the spiked Wigner model, recovering the sparse spike $\boldsymbol{u}$ from the noisy observation $\boldsymbol{Y}$ in (1) naturally leads to the sparse PCA formulation:

$$\max_{\boldsymbol{w}} \boldsymbol{w}^\top \boldsymbol{Y} \boldsymbol{w}, \text{ subject to } \|\boldsymbol{w}\|_2 = 1, \|\boldsymbol{w}\|_0 \leq k. \quad (6)$$

where $k$ is a sparsity parameter. Since $\boldsymbol{u}$ is the leading eigenvector of $\mathbb{E}[\boldsymbol{Y}] = \beta \boldsymbol{u} \boldsymbol{u}^\top$, the solution to (6) provides a natural estimator for $\boldsymbol{u}$.

The truncated power method (Yuan & Zhang, 2013) is an iterative algorithm designed to solve the sparse PCA problem (6). Starting from an initial vector $\boldsymbol{u}^0$, it alternates between power iteration and hard thresholding:

$$\boldsymbol{u}^t = \mathcal{P}_{\mathbb{S}^{d-1}}(\mathcal{H}_k(\boldsymbol{Y} \boldsymbol{u}^{t-1})),$$

where $\mathcal{P}_{\mathbb{S}^{d-1}} : \mathbb{R}^d \setminus \{\boldsymbol{0}\} \to \mathbb{S}^{d-1}$ defined by $\mathcal{P}_{\mathbb{S}^{d-1}}(\boldsymbol{z}) = \boldsymbol{z}/\|\boldsymbol{z}\|_2$, and $\mathcal{H}_k : \mathbb{R}^d \to \mathbb{R}^d$ is the hard thresholding operator that retains the $k$ largest entries (in absolute value) and zeros out the rest. The parameter $k$ denotes the sparsity level used in (6), whereas $s$ denotes the sparsity of the true spike. We assume $k$ is of the same order as the true sparsity $s$ throughout, and set $k = s$ in all experiments.

For computational efficiency, we exploit the sparsity structure. Let $\mathcal{T}^t = \text{supp}(\boldsymbol{u}^t)$ denote the support of iterate $t$. Since $|\mathcal{T}^t| \leq k$, we can rewrite the matrix-vector multiplication as:

$$\boldsymbol{u}^t = \mathcal{P}_{\mathbb{S}^{d-1}}(\mathcal{H}_k(\boldsymbol{Y}_{:,\mathcal{T}^{t-1}} \boldsymbol{u}^{t-1}_{\mathcal{T}^{t-1}})),$$

where $\boldsymbol{Y}_{:,\mathcal{T}}$ denotes the submatrix of $\boldsymbol{Y}$ with columns indexed by $\mathcal{T}$. This reduces the per-iteration complexity from $O(d^2)$ to $O(ds)$.

The sparse PCA problem (6) is highly nonconvex due to the cardinality constraint, resulting in a landscape riddled with local maxima. Like all iterative methods for such problems, the truncated power method's performance hinges on initialization quality—poor starting points can trap the algorithm in suboptimal local maxima or prevent convergence entirely. The choice of initialization thus becomes crucial for achieving good performance.

Yuan & Zhang (2013) provided a sharp characterization of when the truncated power method succeeds: geometric convergence to a near-optimal solution is guaranteed when the initial vector $\boldsymbol{u}^0$ has sufficient correlation with the truth, i.e., $|\langle \boldsymbol{u}^0, \boldsymbol{u} \rangle| \geq c$ for some constant $c > 0$. However, obtaining such initialization is the key challenge—random initialization typically fails in high dimensions, while existing polynomial-time algorithms require $\beta = \widetilde{\Omega}(s)$.

Our column thresholding algorithm provides a simple and effective approach that operates under weaker signal strength conditions in this setting. As shown in Section 2.3, it produces an initialization with two key properties:

- Near-perfect correlation: $|\langle \hat{\boldsymbol{u}}, \boldsymbol{u} \rangle| \geq 1 - \frac{\zeta^2}{2}$ for arbitrarily small $\zeta > 0$.

- Optimal signal strength: It succeeds under $\beta = \Omega(\sqrt{s \log d})$, matching the information-theoretic limit of $\widetilde{\Omega}(\sqrt{s})$.

With this initialization, the truncated power method attains near-optimal estimation accuracy under minimal signal strength requirements—a combination unattainable by either method alone. We formalize the resulting convergence guarantee in Section 3.2, and summarize the full two-stage procedure in Algorithm 3.

Moreover, we emphasize the difference between Yuan & Zhang (2013) and our work. Yuan & Zhang (2013) focused on the iterations itself and its convergence. However, we study the signal-strength threshold at which polynomial-time estimation becomes possible in the spiked Wigner model. Our main contribution is column thresholding, while the truncated power method only serves as its practical refinement but does not sharpen the theoretical guarantee.

---

**Algorithm 3** Truncated Power Method (TPM)

1: **Input:** Matrix $\boldsymbol{Y} \in \mathbb{R}^{d \times d}$, sparsity $s$, parameter $k$
2: $\boldsymbol{u}^0 \leftarrow$ Column Thresholding$(\boldsymbol{Y}, s)$
3: **for** $t = 1, 2, \ldots$ **do**
4:     $\mathcal{T}^{t-1} \leftarrow \text{supp}(\boldsymbol{u}^{t-1})$
5:     $\boldsymbol{z}^t \leftarrow \boldsymbol{Y}_{:,\mathcal{T}^{t-1}} \boldsymbol{u}_{\mathcal{T}^{t-1}}$
6:     $\boldsymbol{u}^t \leftarrow \mathcal{P}_{\mathbb{S}^{d-1}}(\mathcal{H}_k(\boldsymbol{z}^t))$
7: **end for**
8: **Output:** Estimated sparse unit vector $\boldsymbol{u}^t \in \mathbb{R}^d$

---

### 3.1. Computational Complexity

The column thresholding initialization requires $O(d \log d + s^3)$ operations, as detailed in Section 2.2. Each truncated power iteration involves two main steps: (i) a sparse matrix–vector multiplication costing $O(ds)$ operations, and (ii) sorting the resulting vector requiring $O(d \log d)$ operations. Thus, each iteration costs $O(ds + d \log d)$ operations. Since the method converges in $O(\log(1/\epsilon))$ iterations to achieve $\epsilon$-accuracy (Section 3.2), the total refinement cost is $O\big((ds + d \log d) \log(1/\epsilon)\big)$.

### 3.2. Theoretical results

This section establishes a theoretical convergence guarantee for the truncated power method. We show that the iterates contract geometrically toward the true spike: the optimization error decreases by a constant factor at each iteration until it reaches an irreducible statistical floor determined by the noise level. This result formalizes both the rapid algorithmic convergence of the method and the point at which further iterations cannot improve accuracy beyond the inherent statistical limit.

**Theorem 3.1.** *Let $\boldsymbol{u} \in \mathbb{R}^d$ be an s-sparse unit vector and $\boldsymbol{Y} = \beta \boldsymbol{u}\boldsymbol{u}^\top + \boldsymbol{W}$, where $\beta > 0$ and $\boldsymbol{W} \sim \mathrm{GOE}(d)$. Fix any $\zeta \in (0,1)$. There exist universal constants $C_4, C_5 > 0$ such that, if*

$$\beta \geq C_4 \max\big\{\|\boldsymbol{u}\|_\infty^{-1}, \, \zeta^{-1}\big\} \sqrt{s \log d},$$

*then, with probability at least $1 - 1.5d^{-1}$, the sequence $\{\boldsymbol{u}^t\}_{t \geq 1}$ produced by Algorithm 3, initialized with $\boldsymbol{u}^0$ from Algorithm 1 and using parameter $k = C_5 s$, satisfies*

$$\mathrm{dist}(\boldsymbol{u}, \boldsymbol{u}^t) \, \leq \, \eta^t \, \mathrm{dist}(\boldsymbol{u}, \boldsymbol{u}^0) \, + \, h\zeta, \qquad (7)$$

*where $\eta, h \in (0,1)$ are universal constants.*

The proof is provided in Appendix A.7. Theorem 3.1 decomposes the estimation error into two parts. The first term, $\eta^t \mathrm{dist}(\boldsymbol{u}, \boldsymbol{u}^0)$, represents an optimization error

that decays geometrically with iteration count $t$. The second term, $h\zeta$, captures the irreducible statistical error inherent to the problem. Consequently, the truncated power method rapidly eliminates the optimization error—requiring only $O(\log \zeta^{-1})$ iterations to achieve $\zeta$-accuracy—while operating under the optimal signal strength $\beta = \widetilde{\Omega}(\sqrt{s})$ when $\|\boldsymbol{u}\|_\infty = \Omega(1)$.

## 4. Experimental results

We empirically verify that column thresholding satisfies the information-theoretic signal-strength requirement, thereby validating our theoretical guarantees. We further show that TPM outperforms existing methods in estimation accuracy, support recovery, and computational efficiency. Performance is assessed by estimation error (via the metric in (5)) and the F-score (0–1 scale, with 1 indicating perfect recovery). All results are averaged over 200 independent Monte Carlo runs.

### 4.1. Empirical Validation of Optimal Signal Thresholds

We empirically verify that our column thresholding achieves the information-theoretic signal strength requirement for constant estimation error and exact support recovery, thereby validating Theorem 2.2 and Theorem 2.4. In each trial, we construct an $s$-sparse spike $\boldsymbol{u}$ with one entry of magnitude 0.5 and the remaining $s-1$ nonzeros of equal magnitude, normalized so that $\|\boldsymbol{u}\|_2 = 1$. This design ensures two key properties: (i) $\|\boldsymbol{u}\|_\infty$ is constant in $d$ and $s$ (as required by Theorem 2.2), and (ii) every nonzero entry satisfies $|u_i| \geq \theta/\sqrt{s}$ for a universal constant $\theta$ (as required by Theorem 2.4). We then generate $\boldsymbol{Y}$ according to (1).

We conduct experiments across two regimes: (i) varying the dimension $d$ in $\{2000, 5000, 10000, 15000, 20000\}$ with fixed sparsity $s = 20$, and (ii) varying the sparsity $s$ in $\{20, 50, 100, 150, 200\}$ with fixed dimension $d = 10000$. When performance is plotted against the scaled

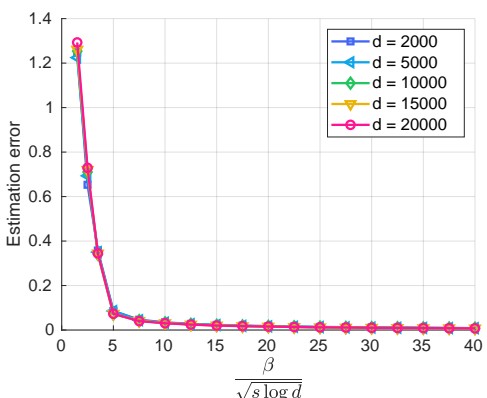
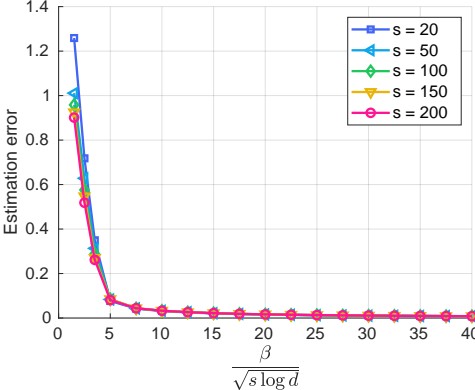

*Figure 2.* Estimation error vs. scaled signal strength for our column thresholding under varying dimensions (left) and sparsities (right).

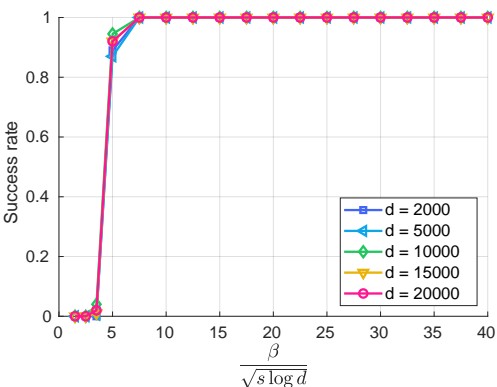 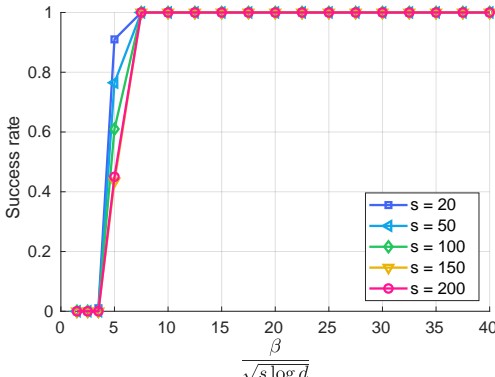

*Figure 3.* Success rate vs. scaled signal strength for our column thresholding under varying dimensions (left) and sparsities (right).

signal strength $\beta/\sqrt{s \log d}$, the curves from different $(d, s)$ collapse, indicating that the phase transition depends only on this scaled quantity, confirming the theoretical scaling.

Figure 2 shows that the estimation-error curves from all tested $(d, s)$ settings collapse once $\beta/\sqrt{s \log d} \geq 10$, essentially independent of the specific values of $d$ and $s$. This indicates that column thresholding succeeds when $\beta \geq C_1 \sqrt{s \log d}$ with a universal constant $C_1 \approx 10$, providing empirical support for the $\Omega(\sqrt{s \log d})$ signal-strength requirement established in Theorem 2.2.

Figure 3 exhibits a sharp phase transition for support recovery around $\beta/\sqrt{s \log d} \approx 7.5$. Below this threshold, perfect recovery is not guaranteed; above it, the success rate rapidly approaches 1 and remains at 1 uniformly across all tested $d$ and $s$. The location and sharpness of this transition are consistent with the predicted $\sqrt{s \log d}$ scaling, and empirically validate the $\Omega(\sqrt{s \log d})$ signal-strength requirement for exact support recovery in Theorem 2.4.

### 4.2. Growing-$s$ experiment and phase-transition behavior

We empirically investigate how the performance of the column–thresholding step evolves as the sparsity level $s$ grows with the dimension $d$. We adopt the same phase-diagram parametrization as in Figure 1:

$$s = \widetilde{\Theta}(d^\phi), \quad \frac{s}{\beta} = \widetilde{\Theta}(d^\psi).$$

For any fixed $\psi$, increasing $\phi$ moves the problem from the "Impossible" region, through the "Hard" region, and eventually into the "Easy" region (see Figure 1).

In this experiment we fix $\psi = 0.2$, for which the phase diagram predicts a transition from "Impossible" to "Hard" at $\phi = 0.4$ and from "Hard" to "Easy" at $\phi = 0.7$. To probe the growing-$s$ behavior along this slice, we consider three

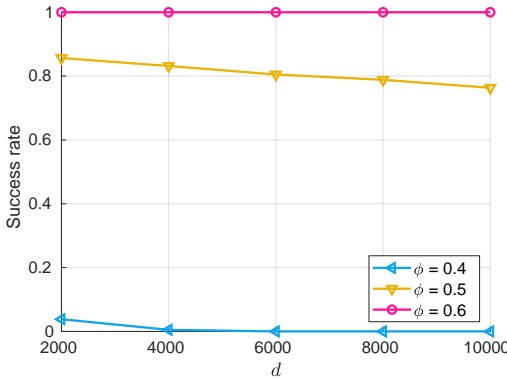

*Figure 4.* Empirical success rate of support recovery by column thresholding as a function of the dimension $d$ for three sparsity scalings: $s = d^{0.4}$, $s = d^{0.5}$, and $s = d^{0.6}$, with $\psi = 0.2$ fixed. The success rate is computed over 500 Monte Carlo trials.

representative sparsity scalings

$$s = d^{0.4}, \quad s = d^{0.5}, \quad s = d^{0.6},$$

corresponding to $\phi = 0.4, 0.5, 0.6$, respectively, and set $\beta = 10sd^{-0.2}$ so that $s/\beta \asymp d^{0.2}$ in all cases. Figure 4 reports the empirical success probability over 500 Monte Carlo trials as a function of $d$ for these three values of $\phi$.

The results in Figure 4 are consistent with the theoretical phase diagram and clearly illustrate the growing-$s$ transition. When $s = d^{0.4}$ (i.e., $\phi = 0.4$, at the boundary between the "Impossible" and "Hard" regions), the success rate remains close to zero across the range of dimensions considered, indicating that the algorithm almost never identifies the true support. When $s = d^{0.6}$ ($\phi = 0.6$, in the "Hard" region), the success probability rises to 1 (exact support recovery in every trial) over the range of $d$ considered.

Overall, these experiments provide a quantitative illustration of the transition in algorithm performance as $s$ grows with $d$, and show that the empirical behavior of the support-recovery step closely matches the theoretically predicted thresholds in $(\phi, \psi)$-space.

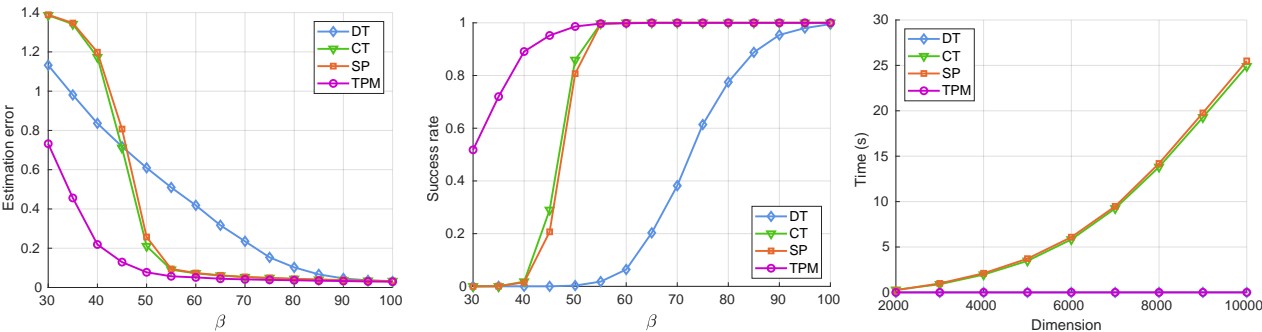

*Figure 5.* Estimation error (left) and support-recovery success rate (middle) as functions of the signal strength $\beta$, and runtime (right) as a function of the dimension $d$.

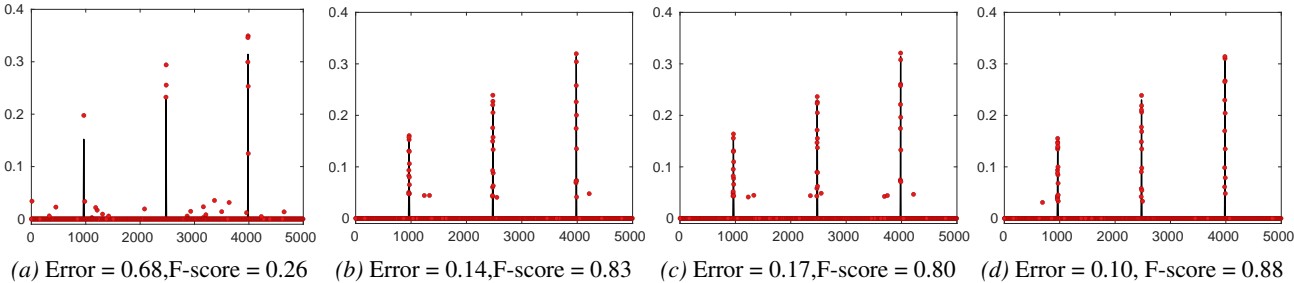

*(a)* Error = 0.68, F-score = 0.26  *(b)* Error = 0.14, F-score = 0.83  *(c)* Error = 0.17, F-score = 0.80  *(d)* Error = 0.10, F-score = 0.88

*Figure 6.* Three-peak benchmark results. True spike (black curve) versus estimated spike (red markers) for four methods: (a) DT, (b) CT, (c) SP, and (d) TPM. The true signal comprises three Beta densities on $[0, 1]$ with dimension $p = 5000$ and signal strength $\beta = 100$.

## 4.3. Statistical and Computational Comparison

We evaluate TPM against three established approaches: diagonal thresholding (DT) (Johnstone & Lu, 2009), co-variance thresholding (CT) (Krauthgamer et al., 2015), and spectral projection (SP) (Brennan et al., 2018). For all experiments, we construct the true spike $u$ with $s$ randomly-located nonzero entries, each taking values $\pm 1/\sqrt{s}$ with equal probability. This balanced spike design, standard in the sparse PCA literature (Krauthgamer et al., 2015), ensures $\|u\|_2 = 1$ while keeping uniform entry magnitudes.

Figure 5 summarizes both statistical performance and computational scalability. In the left and middle panels, we vary the signal strength $\beta$ with $d = 2000$ and $s = 10$ fixed. TPM performs best overall, with its advantage most pronounced in the weak-signal regime: as $\beta$ decreases, CT and SP degrade rapidly while TPM maintains more robust estimation and support recovery. TPM also consistently outperforms DT for all tested values of $\beta$, which aligns with our analysis in Section 2.1 showing that column threshold-ing creates a larger statistical separation between in-support and out-of-support indices than diagonal thresholding. The right panel assesses scalability by varying $d$ from 2000 to 10000 with $\beta = 100$ and $s = 15$ fixed; TPM's runtime scales comparably to DT, while CT and SP are substantially more expensive.

## 4.4. Three-peak Benchmark Evaluation

We evaluate our method on the canonical "three-peak" ex-periment (Johnstone & Lu, 2009), a demanding benchmark for sparse recovery in high dimensions. The true spike $v$ is constructed as a mixture of three Beta densities on $[0, 1]$, producing three pronounced peaks separated by near-zero valleys. This setup rigorously tests an algorithm's ability to localize multiple signal components while suppressing inter-peak noise. The experiment stresses methods' capacity to distinguish true signal peaks from spurious activations. As shown in Figure 6, TPM faithfully recovers all three peaks, achieving lower estimation error and higher F-score than competing methods, which either misestimate the peaks or introduce false detections in the valleys.

## 5. Conclusions

We propose two algorithms for sparse PCA in the spiked Wigner model. Column thresholding runs in polynomial time and achieves the signal scaling $\tilde{\Omega}(\sqrt{s})$ under the non-uniformity condition $\|u\|_\infty = \Omega(1)$, identifying a class of spikes beyond existing planted-clique reductions. Building on this initializer, the truncated power method provably refines the estimate with linear convergence. Experiments validate the theory and show improved estimation, support recovery, and runtime relative to prior methods.

## Acknowledgement

This work was supported by the Hong Kong Research Grant Council (RGC) GRFs 16307325, 16306124, and 16307023, and the Hong Kong RGC Postdoctoral Fellowship Scheme of Project No. PDFS2425-6S05. We would also like to thank the anonymous reviewers for their valuable feedback on the manuscript.

## Impact Statement

This paper aims to advance the theoretical and computational aspects of sparse PCA. We do not foresee any specific ethical concerns arising from this work.

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

# Appendix Contents

## A. Proofs

In Appendix A, we prove the proposition and theorems introduced in Sections 2 and 3. First, we present some auxiliary lemmas in Appendix A.1 and show some technical lemmas in Appendix A.2. Next, we prove Proposition 2.1 in Appendix A.3. Subsequently, we present the proofs for Theorem 2.2, Theorem 2.3 and Theorem 2.4 in Appendix A.4, Appendix A.5 and Appendix A.6, respectively. Finally, we show the proof for Theorem 3.1 in Appendix A.7.

Throughout Appendix A, we define the largest and smallest $\ell$-sparse eigenvalue of a symmetric matrix $\boldsymbol{B} \in \mathbb{R}^{m \times m}$ by

$$\lambda_{\max}(\boldsymbol{B}, \ell) = \max_{\boldsymbol{w} \in \mathbb{R}^m, \|\boldsymbol{w}\|_2 = 1, \|\boldsymbol{w}\|_0 = \ell} \boldsymbol{w}^T \boldsymbol{B} \boldsymbol{w}, \ \lambda_{\min}(\boldsymbol{B}, s) = \min_{\boldsymbol{w} \in \mathbb{R}^m, \|\boldsymbol{w}\|_2 = 1, \|\boldsymbol{w}\|_0 = \ell} \boldsymbol{w}^T \boldsymbol{B} \boldsymbol{w},$$

respectively. Then we define the maximum spectral norm of all $\ell \times \ell$ submatrices of $\boldsymbol{B}$ by

$$\rho(\boldsymbol{B}, l) = \max \left\{ |\lambda_{\max}(\boldsymbol{B}, \ell)|, |\lambda_{\min}(\boldsymbol{B}, \ell)| \right\}. \tag{8}$$

### A.1. Auxiliary Lemmas

The following two lemmas are used to prove the convergence of truncated power method in Algorithm 3, which will be used for the proof of Theorem 3.1 in Appendix A.7.

**Lemma A.1** ((Yuan & Zhang, 2013)). *Let $\boldsymbol{z}$ be the eigenvector with the largest eigenvalue (in absolute value) of a symmetric matrix $\boldsymbol{B}$, and let $\kappa < 1$ be the ratio of the second to the largest eigenvalue in absolute values. Given any $\boldsymbol{y}$ such that $\|\boldsymbol{y}\|_2 = 1$, let $\boldsymbol{y}' = \boldsymbol{B}\boldsymbol{y}/\|\boldsymbol{B}\boldsymbol{y}\|_2$, then*

$$\left| \boldsymbol{z}^\top \boldsymbol{y}' \right| \geq \left| \boldsymbol{z}^\top \boldsymbol{y} \right| \left( 1 + \frac{1}{2} \left( 1 - \kappa^2 \right) \left( 1 - \left| \boldsymbol{z}^\top \boldsymbol{y} \right|^2 \right) \right).$$

**Lemma A.2** ((Yuan & Zhang, 2013)). *Consider $\boldsymbol{y}$ with $\|\boldsymbol{y}\|_0 = \ell$. Consider $\boldsymbol{z}$ and let $\mathcal{F} = \mathrm{supp}(\boldsymbol{z}, \ell')$ be the set of indices with the $\ell'$ largest absolute values in $\boldsymbol{z}$. If $\|\boldsymbol{y}\|_2 = \|\boldsymbol{z}\|_2 = 1$, then*

$$\left| \boldsymbol{y}^\top \boldsymbol{z}_\mathcal{F} \right| \geq \left| \boldsymbol{y}^\top \boldsymbol{z} \right| - \sqrt{\ell/\ell'} \min \left\{ \sqrt{1 - |\boldsymbol{y}^\top \boldsymbol{z}|^2}, (1 + \sqrt{\ell/\ell'}) \left( 1 - \left| \boldsymbol{y}^\top \boldsymbol{z} \right|^2 \right) \right\}.$$

### A.2. Technical Lemmas

In Appendix A.2, we show some technical lemmas that will be used for the proofs of Theorem 2.2 and Theorem 3.1. The first lemma bounds the quantity $\rho(\boldsymbol{W}, \ell)$ defined in (8).

**Lemma A.3.** *For any $r \in (0, 1)$,*

$$\mathbb{P}\left\{ \rho(\boldsymbol{W}, \ell) \leq 3r\beta \right\} \geq 1 - \frac{2}{\sqrt{\pi} r \beta} \left( \frac{9ed}{\ell} \right)^\ell \exp\left( -\frac{r^2 \beta^2}{4} \right). \tag{9}$$

*Proof.* Denote the set of $\ell$-sparse vectors in $\mathbb{R}^d$ by $\mathbb{T}_\ell^d := \{ \boldsymbol{w} : \|\boldsymbol{w}\|_2 = 1, \|\boldsymbol{w}\|_0 = \ell \}$. For any $\delta \in (0, 1)$, there exists a set $\mathcal{N}_\delta \subset \mathbb{T}_\ell^d$ such that for any $\boldsymbol{w} \in \mathbb{T}_\ell^d$, there exists $\boldsymbol{w}_\delta \in \mathcal{N}_\delta$ such that $\mathrm{supp}(\boldsymbol{w}) = \mathrm{supp}(\boldsymbol{w}_\delta)$ and $\|\boldsymbol{w} - \boldsymbol{w}_\delta\|_2 \leq \delta$ and $|\mathcal{N}_\delta| \leq \binom{d}{\ell} (\frac{3}{\delta})^\ell \leq (\frac{3ed}{\delta \ell})^\ell$ (Baraniuk et al., 2008).

From (8), we obtain

$$\rho(\boldsymbol{W}, \ell) = \max_{\substack{\boldsymbol{y}, \boldsymbol{z} \in \mathbb{T}_\ell^d, \\ \mathrm{supp}(\boldsymbol{y}) = \mathrm{supp}(\boldsymbol{z})}} \boldsymbol{y}^\top \boldsymbol{W} \boldsymbol{z} =: \boldsymbol{y}_*^\top \boldsymbol{W} \boldsymbol{z}_*.$$

From the definition of $\mathcal{N}_\delta$, there exists $\boldsymbol{y}_\delta, \boldsymbol{z}_\delta \in \mathcal{N}_\delta$ such that $\mathrm{supp}(\boldsymbol{y}_\delta) = \mathrm{supp}(\boldsymbol{y}_*) = \mathrm{supp}(\boldsymbol{z}_*) = \mathrm{supp}(\boldsymbol{z}_\delta)$, $\|\boldsymbol{y}_* - \boldsymbol{y}_\delta\|_2 \leq \delta$ and $\|\boldsymbol{z}_* - \boldsymbol{z}_\delta\|_2 \leq \delta$. Then we have

$$\boldsymbol{y}_*^\top \boldsymbol{W} \boldsymbol{z}_* = \boldsymbol{y}_*^\top \boldsymbol{W} (\boldsymbol{z}_* - \boldsymbol{z}_\delta) + (\boldsymbol{y}_* - \boldsymbol{y}_\delta)^\top \boldsymbol{W} \boldsymbol{z}_\delta + \boldsymbol{y}_\delta^\top \boldsymbol{W} \boldsymbol{z}_\delta \leq 2\delta \boldsymbol{y}_*^\top \boldsymbol{W} \boldsymbol{z}_* + \boldsymbol{y}_\delta^\top \boldsymbol{W} \boldsymbol{z}_\delta,$$

which implies that

$$\rho(\boldsymbol{W}, \ell) \leq (1 - 2\delta)^{-1} \boldsymbol{y}_\delta^\top \boldsymbol{W} \boldsymbol{z}_\delta \leq (1 - 2\delta)^{-1} \max_{\substack{\boldsymbol{y}, \boldsymbol{z} \in \mathcal{N}_\delta, \\ \mathrm{supp}(\boldsymbol{y}) = \mathrm{supp}(\boldsymbol{z})}} \boldsymbol{y}^\top \boldsymbol{W} \boldsymbol{z}, \tag{10}$$

where the inequalities hold when $1 - 2\delta > 0$.

Now for any $(\boldsymbol{y}, \boldsymbol{z}) \in \mathcal{N}_\delta$, we bound $|\boldsymbol{y}^\top \boldsymbol{W} \boldsymbol{z}|$ as follows. Since $\boldsymbol{W} = \frac{1}{\sqrt{2}}(\boldsymbol{A} + \boldsymbol{A}^\top)$ with some random matrix $\boldsymbol{A} \sim \mathcal{N}(0, 1)^{\otimes d \times d}$, we obtain

$$\boldsymbol{y}^\top \boldsymbol{W} \boldsymbol{z} \sim \mathcal{N}(0, 1 + |\boldsymbol{y}^T \boldsymbol{z}|^2).$$

Therefore, using the tail of a Gaussian variable (Vershynin, 2018), for any $r \in (0, 1)$, it holds that

$$\mathbb{P}\left\{ |\boldsymbol{y}^\top \boldsymbol{W} \boldsymbol{z}| \ge r\beta \right\} \le \sqrt{\frac{2}{\pi}} \frac{\sqrt{1 + |\boldsymbol{y}^T \boldsymbol{z}|^2}}{r\beta} \exp\left( -\frac{r^2 \beta^2}{2(1 + |\boldsymbol{y}^T \boldsymbol{z}|^2)} \right) \le \frac{2}{\sqrt{\pi} r\beta} \exp\left( -\frac{r^2 \beta^2}{4} \right),$$

where the last inequality we use $\|\boldsymbol{y}\|_2 = \|\boldsymbol{z}\|_2 = 1$.

By taking union bounds for all $(\boldsymbol{y}, \boldsymbol{z}) \in \mathcal{N}_\delta$, we obtain that

$$\mathbb{P}\left\{ \max_{\substack{\boldsymbol{y}, \boldsymbol{z} \in \mathcal{N}_\delta, \\ \mathrm{supp}(\boldsymbol{y}) = \mathrm{supp}(\boldsymbol{z})}} |\boldsymbol{y}^\top \boldsymbol{W} \boldsymbol{z}| \le r\beta \right\} \ge 1 - \frac{2}{\sqrt{\pi} r\beta} \left( \frac{3ed}{\delta\ell} \right)^\ell \exp\left( -\frac{r^2 \beta^2}{4} \right).$$

Setting $\delta = \frac{1}{3}$ together with (10) leads to (9). $\qquad\square$

The next lemma bounds the error between $\boldsymbol{u}$ and the $\ell$-sparse largest eigenvector of $\boldsymbol{Y}$.

**Lemma A.4.** *Let $\Lambda \subset [d]$ be such that $\Lambda \bigcup \mathcal{T} \ne \emptyset$ and $|\Lambda| = \ell$. Let $\boldsymbol{w}$ be the largest eigenvector of $\boldsymbol{Y}_\Lambda$ with $\|\boldsymbol{w}\|_2 = 1$. If $\rho(\boldsymbol{W}, \ell) < \frac{\beta}{2} \|\boldsymbol{u}_\Lambda\|_2^2$, then we have*

$$\mathrm{dist}(\boldsymbol{w}, \boldsymbol{u}_\Lambda)^2 \le \|\boldsymbol{u}_\Lambda\|_2^2 + 1 - 2 \frac{\|\boldsymbol{u}_\Lambda\|_2}{\sqrt{1 + \frac{\rho(\boldsymbol{W}, \ell)^2}{\left( \beta \|\boldsymbol{u}_\Lambda\|_2^2 - 2\rho(\boldsymbol{W}, \ell) \right)^2}}}.$$

*Proof.* Denote $\bar{\lambda}$ the largest eigenvalue of $\boldsymbol{Y}_\Lambda$, i.e. $\bar{\lambda} = \lambda_1(\boldsymbol{Y}_\Lambda)$. Recall that $\boldsymbol{Y} = \beta \boldsymbol{u} \boldsymbol{u}^\top + \boldsymbol{W}$ and $\mathbb{E}[\boldsymbol{Y}] = \beta \boldsymbol{u} \boldsymbol{u}^\top$. Using Weyl's inequality (Horn & Johnson, 2012), it holds that

$$\bar{\lambda} \ge \lambda_1(\mathbb{E}[\boldsymbol{Y}_\Lambda]) + \lambda_n(\boldsymbol{W}_\Lambda) \ge \beta \|\boldsymbol{u}_\Lambda\|_2^2 - \rho(\boldsymbol{W}, \ell), \tag{11}$$

where the last inequality holds by $\lambda_n(\boldsymbol{W}_\Lambda) \ge -\rho(\boldsymbol{W}, \ell)$ from the definition. Similarly, we have for all $i \ge 2$,

$$\begin{aligned} |\lambda_i(\boldsymbol{Y}_\Lambda)| &\le |\lambda_i(\mathbb{E}[\boldsymbol{Y}_\Lambda])| + |\lambda_i(\boldsymbol{Y}_\Lambda) - \lambda_i(\mathbb{E}[\boldsymbol{Y}_\Lambda])| \\ &= \max\{|\lambda_1(\boldsymbol{W}_\Lambda)|, |\lambda_n(\boldsymbol{W}_\Lambda)|\} \le \rho(\boldsymbol{W}, \ell). \end{aligned} \tag{12}$$

Notice $\|\boldsymbol{w}\|_2 = 1$ but $\|\boldsymbol{u}_\Lambda\|_2 \le 1$. We divide $\boldsymbol{w}$ as

$$\boldsymbol{w} = a_1 \frac{\boldsymbol{u}_\Lambda}{\|\boldsymbol{u}_\Lambda\|_2} + a_2 \boldsymbol{y}$$

with $\boldsymbol{u}_\Lambda^\top \boldsymbol{y} = 0$, $\|\boldsymbol{y}\|_2 = 1$ and $a_1^2 + a_2^2 = 1$. Then we have $\mathrm{supp}(\boldsymbol{y}) \subset \Lambda$, and

$$\bar{\lambda} a_1 \frac{\boldsymbol{u}_\Lambda}{\|\boldsymbol{u}_\Lambda\|_2} + \bar{\lambda} a_2 \boldsymbol{y} = \bar{\lambda} \boldsymbol{w} = \boldsymbol{Y}_\Lambda \boldsymbol{w} = a_1 \frac{\boldsymbol{Y}_\Lambda \boldsymbol{u}_\Lambda}{\|\boldsymbol{u}_\Lambda\|_2} + a_2 \boldsymbol{Y}_\Lambda \boldsymbol{y}.$$

By taking the inner product with $\boldsymbol{y}$, we obtain

$$\bar{\lambda} a_2 = a_1 \frac{\boldsymbol{y}^\top \boldsymbol{Y}_\Lambda \boldsymbol{u}_\Lambda}{\|\boldsymbol{u}_\Lambda\|_2} + a_2 \boldsymbol{y}^\top \boldsymbol{Y}_\Lambda \boldsymbol{y}.$$

Since $\boldsymbol{u}_\Lambda$ is the eigenvector of $\mathbb{E}[\boldsymbol{Y}_\Lambda]$ and $\boldsymbol{u}_\Lambda^\top \boldsymbol{y} = 0$, we have $\boldsymbol{y}^\top \mathbb{E}[\boldsymbol{Y}_\Lambda] \boldsymbol{u}_\Lambda = 0$. This leads to

$$|a_2| = |a_1| \frac{\left| \boldsymbol{y}^\top (\boldsymbol{W}_\Lambda + \mathbb{E}[\boldsymbol{Y}_\Lambda]) \frac{\boldsymbol{u}_\Lambda}{\|\boldsymbol{u}_\Lambda\|_2} \right|}{\left| \bar{\lambda} - \boldsymbol{y}^\top \boldsymbol{Y}_\Lambda \boldsymbol{y} \right|} = |a_1| \frac{\left| \boldsymbol{y}^\top \boldsymbol{W}_\Lambda \frac{\boldsymbol{u}_\Lambda}{\|\boldsymbol{u}_\Lambda\|_2} \right|}{\left| \bar{\lambda} - \boldsymbol{y}^\top \boldsymbol{Y}_\Lambda \boldsymbol{y} \right|}.$$

Since $\operatorname{supp}(\boldsymbol{y}) \subset \Lambda$, we have $\left| \boldsymbol{y}^\top \boldsymbol{W}_\Lambda \frac{\boldsymbol{u}_\Lambda}{\|\boldsymbol{u}_\Lambda\|_2} \right| \leq \rho(\boldsymbol{W}, \ell)$. Moreover, since $\boldsymbol{y}$ is perpendicular to $\boldsymbol{u}_\Lambda$, from (12) we have

$$\left| \boldsymbol{y}^\top \boldsymbol{Y}_\Lambda \boldsymbol{y} \right| \leq \max_{i \geq 2} |\lambda_i(\boldsymbol{Y}_\Lambda)| \leq \rho(\boldsymbol{W}, \ell).$$

So from (11) and $\rho(\boldsymbol{W}, \ell) < \frac{\beta}{2} \|\boldsymbol{u}_\Lambda\|_2^2$, we have

$$\frac{|a_2|}{|a_1|} = \frac{\left| \boldsymbol{y}^\top \boldsymbol{W}_\Lambda \frac{\boldsymbol{u}_\Lambda}{\|\boldsymbol{u}_\Lambda\|_2} \right|}{\left| \bar{\lambda} - \boldsymbol{y}^\top \boldsymbol{Y}_\Lambda \boldsymbol{y} \right|} \leq \frac{\rho(\boldsymbol{W}, \ell)}{\beta \|\boldsymbol{u}_\Lambda\|_2^2 - 2\rho(\boldsymbol{W}, \ell)}.$$

Then, since $a_1^2 + a_2^2 = 1$, we have

$$a_1^2 \geq \frac{1}{1 + \frac{\rho(\boldsymbol{W}, \ell)^2}{\left( \beta \|\boldsymbol{u}_\Lambda\|_2^2 - 2\rho(\boldsymbol{W}, \ell) \right)^2}},$$

which implies that

$$\begin{aligned}
\operatorname{dist}(\boldsymbol{w}, \boldsymbol{u}_\Lambda)^2 &= \min \left\{ \|\boldsymbol{u}_\Lambda - \boldsymbol{w}\|_2^2, \|\boldsymbol{u}_\Lambda + \boldsymbol{w}\|_2^2 \right\} \\
&= \|\boldsymbol{u}_\Lambda\|_2^2 + 1 - 2|a_1| \cdot \|\boldsymbol{u}_\Lambda\|_2 \\
&\leq \|\boldsymbol{u}_\Lambda\|_2^2 + 1 - 2\frac{\|\boldsymbol{u}_\Lambda\|_2}{\sqrt{1 + \frac{\rho(\boldsymbol{W}, \ell)^2}{\left( \beta \|\boldsymbol{u}_\Lambda\|_2^2 - 2\rho(\boldsymbol{W}, \ell) \right)^2}}}.
\end{aligned}$$

$\square$

## A.3. Proof of Proposition 2.1

*Proof of Proposition 2.1.* Recall that $\boldsymbol{Y} = \lambda \boldsymbol{u}\boldsymbol{u}^\top + \boldsymbol{W}$ and $\mathbb{E}[\boldsymbol{Y}] = \lambda \boldsymbol{u}\boldsymbol{u}^\top$. For any $i \in \mathcal{T}$ and any $i' \in \mathcal{T}^c$, using (2)(3), we obtain

$$\begin{aligned}
Y_{ii} &\geq \left( \mathbb{E}[\boldsymbol{Y}] \right)_{ii} - \left| \left( \mathbb{E}[\boldsymbol{Y}] \right)_{ii} - |Y_{ii}| \right| \geq \beta |u_i|^2 - \frac{1}{2} g_{\mathrm{diag}}, \\
Y_{i'i'} &\leq \left| \left( \mathbb{E}[\boldsymbol{Y}] \right)_{i'i'} \right| + \left| \left( \mathbb{E}[\boldsymbol{Y}] \right)_{i'i'} - |Y_{i'i'}| \right| \leq \frac{1}{2} g_{\mathrm{diag}}.
\end{aligned}$$

Following from the fact that $g_{\mathrm{diag}} = \beta \cdot \min_{i \in \mathcal{T}} |u_i|^2$, one has $Y_{ii} \geq Y_{i'i'}$. $\square$

## A.4. Proof of Theorem 2.2

In Appendix A.4, we prove Theorem 2.2 in three steps. First, we prove that the index $i_0$ chosen in Algorithm 1 satisfies $|u_{i_0}| \geq \frac{\|\boldsymbol{u}\|_\infty}{2}$ with high probability. Second, we show that $\widehat{\mathcal{T}}$ chosen in Algorithm 1 contains the indices of most of the larger nonzero entries of $\boldsymbol{u}$ with high probability. Finally, we put everything together.

*Step 1: Estimating $|u_{i_0}|$.* Recall that $i_0 = \arg\max_{i \in [d]} Y_{ii}$.

**Lemma A.5.** *If $\beta$ satisfies*

$$\beta \geq \frac{32\sqrt{2}}{3} \|\boldsymbol{u}\|_\infty^{-2} \sqrt{\log d},$$

*with probability exceeding $1 - \frac{1}{4\sqrt{2\pi \log 2}} d^{-1}$, $|u_{i_0}| \geq \frac{\|\boldsymbol{u}\|_\infty}{2}$.*

*Proof.* From (1)(4), for any $i \in [d]$, we obtain $Y_{ii} = \beta |u_i|^2 + W_{ii}$ and $\mathbb{E}[Y_{ii}] = \beta |u_i|^2$.

Firstly, we consider $Y_{i_* i_*}$, where $i_*$ satisfies $|u_{i_*}| = \|\boldsymbol{u}\|_\infty$. Since $W_{i_* i_*} \sim \mathcal{N}(0, 2)$, using the tail of a Gaussian variable (Vershynin, 2018), it holds that, for any $\epsilon_1 > 0$,

$$\mathbb{P}\left\{ Y_{i_* i_*} - \beta \|\boldsymbol{u}\|_\infty^2 \leq -\epsilon_1 \right\} = \mathbb{P}\left\{ W_{i_* i_*} \leq -\epsilon_1 \right\} \leq \frac{1}{\sqrt{\pi}\epsilon_1} \exp\left( -\frac{\epsilon_1^2}{4} \right). \tag{13}$$

Secondly, we consider $\mathcal{T}_1 := \left\{ i \in [d] : |u_i| < \frac{\|\boldsymbol{u}\|_\infty}{2} \right\}$. Since $W_{ii} \sim \mathcal{N}(0, 2)$, taking union bound and using the tail of a Gaussian variable (Vershynin, 2018), we have, for any $\epsilon_2 > 0$,

$$
\begin{aligned}
\mathbb{P}\left\{ \max_{i \in \mathcal{T}_1} \left( Y_{ii} - \beta |u_i|^2 \right) \geq \epsilon_2 \right\} &\leq (d-1)\mathbb{P}\left\{ W_{ii} \geq \epsilon_2 \text{ for some } i \in \mathcal{T}_1 \right\} \\
&\leq \frac{d-1}{\sqrt{\pi}\epsilon_2} \exp\left( -\frac{\epsilon_2^2}{4} \right).
\end{aligned}
\tag{14}
$$

Now we combine (13)(14) and set $\epsilon_1 = \epsilon_2 = \frac{3}{8}\beta\|\boldsymbol{u}\|_\infty^2$. The complementary events in (13)(14) are

$$
Y_{i_* i_*} \geq \beta\|\boldsymbol{u}\|_\infty^2 - \frac{3}{8}\beta\|\boldsymbol{u}\|_\infty^2,
$$

$$
\max_{i \in \mathcal{T}_1} \left( Y_{ii} - \beta|u_i|^2 \right) \leq \frac{3}{8}\beta\|\boldsymbol{u}\|_\infty^2,
$$

which leads to

$$
\max_{i \in \mathcal{T}_1} Y_{ii} < \frac{3}{8}\beta\|\boldsymbol{u}\|_\infty^2 + \beta\left( \frac{\|\boldsymbol{u}\|_\infty}{2} \right)^2 = \beta\|\boldsymbol{u}\|_\infty^2 - \frac{3}{8}\beta\|\boldsymbol{u}\|_\infty^2 < Y_{i_* i_*} \leq Y_{i_0 i_0},
$$

where we use the definition of $\mathcal{T}_1$ in the first inequality. It follows that $i_0 \notin \mathcal{T}_1$, i.e. $|u_{i_0}| \geq \frac{\|\boldsymbol{u}\|_\infty}{2}$. Therefore, using (13)(14), we obtain

$$
\mathbb{P}\left\{ |u_{i_0}| \geq \frac{\|\boldsymbol{u}\|_\infty}{2} \right\} \geq 1 - \frac{8d}{3\sqrt{\pi}\beta\|\boldsymbol{u}\|_\infty^2} \exp\left( -\frac{9\beta^2\|\boldsymbol{u}\|_\infty^4}{256} \right),
\tag{15}
$$

which leads to the desired result with the condition of $\beta$. $\qquad\square$

*Step 2: Estimating $\|\boldsymbol{u}_{\widehat{\mathcal{T}}}\|_2$.* For any $\zeta \in (0, 1]$, we define $\mathcal{T}_\zeta^- := \left\{ i \in \mathcal{T} : |u_i| < \frac{\zeta}{2\sqrt{s}} \right\}$ and $\mathcal{T}_\zeta^+ = \mathcal{T} \setminus \mathcal{T}_\zeta^-$. Then we have $\|\boldsymbol{u}_{\mathcal{T}_\zeta^-}\|_2^2 < \frac{\zeta^2}{4s} \cdot s = \frac{\zeta^2}{4}$ and $\|\boldsymbol{u}_{\mathcal{T}_\zeta^+}\|_2^2 \geq 1 - \frac{\zeta^2}{4}$. Since $\|\boldsymbol{u}\|_\infty \geq \frac{1}{\sqrt{s}}$, Lemma A.5 implies that $|u_{i_0}| \geq \frac{1}{2\sqrt{s}} \geq \frac{\zeta}{2\sqrt{s}}$ with high probability, and thus $i_0 \in \mathcal{T}_\zeta^+$. The following lemma shows that $\mathcal{T}_\zeta^+ \subset \widehat{\mathcal{T}}$ with high probability, where $\widehat{\mathcal{T}}$ is chosen in Algorithm 1.

**Lemma A.6.** *For any $\zeta \in (0, 1]$, if $\beta$ satisfies*

$$
\beta \geq 16\zeta^{-1}\|\boldsymbol{u}\|_\infty^{-1}\sqrt{s(\log d + 2\log s)},
$$

*with probability exceeding $1 - \frac{7 + 3\sqrt{2}}{6\sqrt{\pi}\log 2}d^{-1}$, $\mathcal{T}_\zeta^+ \subset \widehat{\mathcal{T}}$.*

*Proof.* It suffices to show that with high probability,

$$
\min_{i \in \mathcal{T}_\zeta^+} |Y_{i, i_0}| > \max_{i \in \mathcal{T}^c} |Y_{i, i_0}|.
$$

To prove this, first, we show that for any $l \in \mathcal{T}_2$, where $\mathcal{T}_2 := \left\{ i \in \mathcal{T} : |u_i| \geq \frac{\|\boldsymbol{u}\|_\infty}{2} \right\}$,

$$
\min_{i \in \mathcal{T}_\zeta^+} |Y_{il}| > \max_{i \in \mathcal{T}^c} |Y_{il}|,
$$

which needs to bound $|Y_{il}|$ and $\left| Y_{il} - \mathbb{E}[Y_{il}] \right|$ for all $i \in \mathcal{T}^c$ and $i \in \mathcal{T}_\zeta^+$.

For any $l \in \mathcal{T}_2$, we first consider $\max_{i \in \mathcal{T}^c} |Y_{il}|$. From (4), for any $i \in \mathcal{T}^c$, we have $\mathbb{E}[Y_{il}] = 0$, and thus $Y_{il} = W_{il}$ by (1). Since $W_{il} \sim \mathcal{N}(0, 1)$, by taking union bound and using the tail of a Gaussian variable (Vershynin, 2018), it holds that, for any $\epsilon_3 > 0$,

$$
\mathbb{P}\left\{ \max_{i \in \mathcal{T}^c} |Y_{il}| \geq \epsilon_3 \right\} \leq \frac{\sqrt{2}(d-s)}{\sqrt{\pi}\epsilon_3} \exp\left( -\frac{\epsilon_3^2}{2} \right).
\tag{16}
$$

Second, we consider $\min_{i \in \mathcal{T}_\zeta^+} \left| \mathbb{E}[Y_{il}] \right|$. From (4), $\mathbb{E}[Y_{il}] = \beta u_i u_l$ for any $i \in \mathcal{T}_\zeta^+$. Then, from the definition of $\mathcal{T}_2$ and $\mathcal{T}_\zeta^+$, we obtain

$$\min_{i \in \mathcal{T}_\zeta^+} \left| \mathbb{E}[Y_{il}] \right| \geq \frac{\zeta \beta \|\boldsymbol{u}\|_\infty}{4\sqrt{s}}. \tag{17}$$

Third, we estimate $\max_{i \in \mathcal{T}} \left| Y_{il} - \mathbb{E}[Y_{il}] \right|$. By (1), $Y_{il} = \beta u_i u_l + W_{il}$. Since $W_{il} \sim \mathcal{N}(0, 1)$ if $i \neq l$ or $W_{il} \sim \mathcal{N}(0, 2)$ if $i = l$, by taking union bound and using the tail of a Gaussian variable (Vershynin, 2018), we have, for any $\epsilon_4 > 0$,

$$\mathbb{P}\left\{ \max_{i \in \mathcal{T}} \left| Y_{il} - \mathbb{E}[Y_{il}] \right| \geq \epsilon_4 \right\} \leq \frac{2s}{\sqrt{\pi} \epsilon_4} \exp\left( -\frac{\epsilon_4^2}{4} \right). \tag{18}$$

Now we combine (16)(18) and set $\epsilon_3 = \epsilon_4 = \frac{\zeta \beta \|\boldsymbol{u}\|_\infty}{8\sqrt{s}}$. The complementary event in (16) is

$$\max_{i \in \mathcal{T}^c} |Y_{il}| \leq \frac{\zeta \beta \|\boldsymbol{u}\|_\infty}{8\sqrt{s}}.$$

Moreover, (17) and the complementary event in (18) lead to

$$|Y_{il}| > \left| \left| \mathbb{E}[Y_{il}] \right| - \left| Y_{il} - \mathbb{E}[Y_{i,i_0}] \right| \right| > \frac{\zeta \beta \|\boldsymbol{u}\|_\infty}{4\sqrt{s}} - \frac{\zeta \beta \|\boldsymbol{u}\|_\infty}{8\sqrt{s}} = \frac{\zeta \beta \|\boldsymbol{u}\|_\infty}{8\sqrt{s}}, \forall i \in \mathcal{T}_\zeta^+.$$

These two inequalities implies that $\min_{i \in \mathcal{T}_\zeta^+} |Y_{il}| > \max_{i \in \mathcal{T}^c} |Y_{il}|$ for any $l \in \mathcal{T}_2$.

Finally, by taking union bound and using (15)(16)(18), we obtain

$$\begin{aligned}
&\mathbb{P}\left\{ \mathcal{T}_\zeta^+ \subset \widehat{\mathcal{T}} \right\} \\
&= \sum_{l \in \mathcal{T}_2} \mathbb{P}\left\{ \min_{i \in \mathcal{T}_\zeta^+} |Y_{il}| > \max_{i \in \mathcal{T}^c} |Y_{il}|, i_0 = l \right\} \\
&\geq \sum_{l \in \mathcal{T}_2} (1 - \mathbb{P}\left\{ \min_{i \in \mathcal{T}_\zeta^+} |Y_{il}| \leq \max_{i \in \mathcal{T}^c} |Y_{il}| \right\} - \mathbb{P}\{i_0 \neq l\}) \\
&\geq \sum_{l \in \mathcal{T}_2} (\mathbb{P}\{i_0 = l\} - \mathbb{P}\left\{ \min_{i \in \mathcal{T}_\zeta^+} |Y_{il}| \leq \max_{i \in \mathcal{T}^c} |Y_{il}| \right\}) \\
&\geq 1 - \frac{8d}{3\sqrt{\pi} \beta \|\boldsymbol{u}\|_\infty^2} \exp\left( -\frac{9\beta^2 \|\boldsymbol{u}\|_\infty^4}{256} \right) - \frac{8\sqrt{2s} s(d-s)}{\sqrt{\pi} \zeta \beta \|\boldsymbol{u}\|_\infty} \exp\left( -\frac{\zeta^2 \beta^2 \|\boldsymbol{u}\|_\infty^2}{128s} \right) \\
&\quad - \frac{16\sqrt{s} s^2}{\sqrt{\pi} \zeta \beta \|\boldsymbol{u}\|_\infty} \exp\left( -\frac{\zeta^2 \beta^2 \|\boldsymbol{u}\|_\infty^2}{256s} \right).
\end{aligned} \tag{19}$$

Since $\zeta \in (0, 1]$, (19) leads to the desired result with the condition of $\beta$. $\qquad \square$

*Step 3: Putting everything together.* Now we estimate $\text{dist}(\hat{\boldsymbol{u}}, \boldsymbol{u})$ and prove Theorem 2.2.

*Proof of Theorem 2.2.* For simplicity, we denote $\rho = \rho(\boldsymbol{W}, s)$. By applying Lemma A.3 with $r = \frac{1}{16}\zeta, \ell = s$ and Lemma A.6, if $\beta$ satisfies

$$\beta \geq \max\left\{ 16\zeta^{-1} \|\boldsymbol{u}\|_\infty^{-1} \sqrt{s(\log d + 2\log s)}, 32\zeta^{-1} \sqrt{\log d + s\log(\frac{9ed}{s})} \right\}, \tag{20}$$

then we have

$$\mathbb{P}\left\{ \rho \leq \frac{3}{16}\zeta\beta, i_0 \in \mathcal{T}_\zeta^+ \subset \widehat{\mathcal{T}} \right\} \geq 1 - \frac{1}{\sqrt{\pi \log(18e)}} d^{-1} - \frac{7 + 3\sqrt{2}}{6\sqrt{\pi \log 2}} d^{-1} > 1 - 1.5558 d^{-1}. \tag{21}$$

Under the event in (21), we estimate $\mathrm{dist}(\hat{\boldsymbol{u}}, \boldsymbol{u})$. Since $\mathrm{supp}(\hat{\boldsymbol{u}}) = \widehat{\mathcal{T}}$, we have

$$\mathrm{dist}(\hat{\boldsymbol{u}}, \boldsymbol{u})^2 = \mathrm{dist}(\hat{\boldsymbol{u}}, \boldsymbol{u}_{\widehat{\mathcal{T}}})^2 + \|\boldsymbol{u}_{\widehat{\mathcal{T}}^c}\|_2^2. \tag{22}$$

Firstly, we estimate $\|\boldsymbol{u}_{\widehat{\mathcal{T}}^c}\|_2^2$. Since $\widehat{\mathcal{T}}^c \subset (\mathcal{T} \setminus \mathcal{T}_\zeta^-)^c = \mathcal{T}_\zeta^- \bigcup \mathcal{T}^c$, we have

$$\|\boldsymbol{u}_{\widehat{\mathcal{T}}^c}\|_2^2 \le \|\boldsymbol{u}_{\mathcal{T}_\zeta^-}\|_2^2 + \|\boldsymbol{u}_{\mathcal{T}^c}\|_2^2 < \frac{\zeta^2}{4} < \frac{1}{4}, \quad \|\boldsymbol{u}_{\widehat{\mathcal{T}}}\|_2^2 > 1 - \frac{\zeta^2}{4} > \frac{3}{4}.$$

Secondly, we estimate $\mathrm{dist}(\hat{\boldsymbol{u}}, \boldsymbol{u}_{\widehat{\mathcal{T}}})^2$. Applying Lemma A.4 with $\Lambda = \widehat{\mathcal{T}}$ and $\ell = s$, we obtain

$$\mathrm{dist}(\hat{\boldsymbol{u}}, \boldsymbol{u}_{\widehat{\mathcal{T}}})^2 \le \|\boldsymbol{u}_{\widehat{\mathcal{T}}}\|_2 + 1 - 2 \frac{\|\boldsymbol{u}_{\widehat{\mathcal{T}}}\|_2}{\sqrt{1 + \frac{\rho^2}{(\beta\|\boldsymbol{u}_{\widehat{\mathcal{T}}}\|_2^2 - 2\rho)^2}}} \le \|\boldsymbol{u}_{\widehat{\mathcal{T}}}\|_2 + 1 - 2 \frac{\|\boldsymbol{u}_{\widehat{\mathcal{T}}}\|_2}{\sqrt{1 + \frac{\rho^2}{(\frac{3}{4}\beta - 2\rho)^2}}},$$

where the last inequality holds since $\|\boldsymbol{u}_{\widehat{\mathcal{T}}}\|_2^2 > \frac{3}{4}$. Therefore, using Lemma A.4 and $\|\boldsymbol{u}_{\widehat{\mathcal{T}}^c}\|_2^2 \le \frac{\zeta^2}{4}$, we have

$$\mathrm{dist}(\hat{\boldsymbol{u}}, \boldsymbol{u}_{\widehat{\mathcal{T}}})^2 \le \max\left\{ 2 - \frac{\zeta^2}{4} - \frac{2\sqrt{1 - \frac{\zeta^2}{4}}}{\sqrt{1 + \frac{\rho^2}{(\frac{3}{4}\beta - 2\rho)^2}}}, 2 - \frac{2}{\sqrt{1 + \frac{\rho^2}{(\frac{3}{4}\beta - 2\rho)^2}}} \right\}$$

$$\le \max\left\{ 2 - \frac{\zeta^2}{4} - 2\frac{1 - \frac{\zeta^2}{4}}{1 + \frac{\rho^2}{(\frac{3}{4}\beta - 2\rho)^2}}, 2 - 2\frac{1}{1 + \frac{\rho^2}{(\frac{3}{4}\beta - 2\rho)^2}} \right\}$$

$$= \max\left\{ \frac{\frac{\zeta^2}{4}(\frac{3}{4}\beta - 2\rho)^2 + (2 - \frac{\zeta^2}{4})\rho^2}{(\frac{3}{4}\beta - 2\rho)^2 + \rho^2}, \frac{2\rho^2}{(\frac{3}{4}\beta - 2\rho)^2 + \rho^2} \right\}$$

$$\le \frac{\zeta^2}{4} + \frac{2\rho^2}{(\frac{3}{4}\beta - 2\rho)^2 + \rho^2}.$$

It follows from (22) and $\rho \le \frac{3}{16}\zeta\beta$ that

$$\mathrm{dist}(\hat{\boldsymbol{u}}, \boldsymbol{u})^2 \le \frac{\zeta^2}{2} + \frac{\zeta^2}{2} = \zeta^2,$$

completing the proof. $\square$

*Remark* A.7. From (20), a sufficient condition for the constant $C_1$ in Theorem 2.2 is

$$C_1 \ge \max\left\{ 16\sqrt{3}, 32\sqrt{2 + \log_2(9\mathrm{e})} \right\} = 32\sqrt{2 + \log_2(9\mathrm{e})}.$$

## A.5. Proof of Theorem 2.3

Since Algorithm 1 and Algorithm 2 have the same step for support estimation, we use some results and techniques in Section A.4 to prove Theorem 2.3. Specifically, it requires Lemma A.5 and Lemma A.6, which show that with high probability, $|u_{i_0}| \ge \frac{\|\boldsymbol{u}\|_\infty}{2}$ and $\mathcal{T}_\zeta^+ \subset \widehat{\mathcal{T}}$. Recall that $\hat{\boldsymbol{u}}_{\mathrm{nv}} = \boldsymbol{Y}_{\widehat{\mathcal{T}}, i_0} / \|\boldsymbol{Y}_{\widehat{\mathcal{T}}, i_0}\|_2$ and $\mathcal{T}_\zeta^+ = \left\{ i \in \mathcal{T} : |u_i| \ge \frac{\zeta}{2\sqrt{s}} \right\}$.

*Proof.* From (5), similar to (22), we have

$$\mathrm{dist}(\hat{\boldsymbol{u}}_{\mathrm{nv}}, \boldsymbol{u})^2 = \mathrm{dist}(\hat{\boldsymbol{u}}_{\mathrm{nv}}, \boldsymbol{u}_{\widehat{\mathcal{T}}})^2 + \|\boldsymbol{u}_{\widehat{\mathcal{T}}^c}\|_2^2. \tag{23}$$

Recall that $\|\boldsymbol{u}_{\widehat{\mathcal{T}}^c}\|_2^2 \le \frac{\zeta^2}{4}$ and $\|\boldsymbol{u}_{\widehat{\mathcal{T}}}\|_2^2 \ge 1 - \frac{\zeta^2}{4}$ from the proof of Theorem 2.2, hence we only need to estimate $\mathrm{dist}(\hat{\boldsymbol{u}}_{\mathrm{nv}}, \boldsymbol{u}_{\widehat{\mathcal{T}}})^2$.

From the definition of $\hat{\boldsymbol{u}}_{\mathrm{nv}}$, we obtain

$$\mathrm{dist}(\hat{\boldsymbol{u}}_{\mathrm{nv}}, \boldsymbol{u}_{\widehat{\mathcal{T}}})^2 = \mathrm{dist}(\frac{\boldsymbol{Y}_{\widehat{\mathcal{T}}, i_0}}{\|\boldsymbol{Y}_{\widehat{\mathcal{T}}, i_0}\|_2}, \boldsymbol{u}_{\widehat{\mathcal{T}}})^2,$$

To handle the randomness of $i_0$. Similarly to Lemma A.6, we estimate

$$\text{dist}(\frac{\boldsymbol{Y}_{\widehat{\mathcal{T}},l}}{\|\boldsymbol{Y}_{\widehat{\mathcal{T}},l}\|_2}, \boldsymbol{u}_{\widehat{\mathcal{T}}})^2 \tag{24}$$

for any $l \in \mathcal{T}_2 = \left\{ i \in \mathcal{T} : |u_i| \geq \frac{\|\boldsymbol{u}\|_\infty}{2} \right\}$. Without loss of generality, we assume $u_l > 0$ and consider

$$\|\frac{\boldsymbol{Y}_{\widehat{\mathcal{T}},l}}{\|\boldsymbol{Y}_{\widehat{\mathcal{T}},l}\|_2} - \boldsymbol{u}_{\widehat{\mathcal{T}}}\|_2^2, \tag{25}$$

which is an upper bound of (24).

We begin with a simplification of (25):

$$
\begin{aligned}
\|\frac{\boldsymbol{Y}_{\widehat{\mathcal{T}},l}}{\|\boldsymbol{Y}_{\widehat{\mathcal{T}},l}\|_2} - \boldsymbol{u}_{\widehat{\mathcal{T}}}\|_2^2 &= \sum_{i \in \widehat{\mathcal{T}}} \left| \frac{Y_{il}}{\|\boldsymbol{Y}_{\widehat{\mathcal{T}},l}\|_2} - u_i \right|^2 \\
&= \sum_{i \in \widehat{\mathcal{T}}} \frac{\left| Y_{il} - \beta u_i u_l + u_i \left( \beta u_l - \|\boldsymbol{Y}_{\widehat{\mathcal{T}},l}\|_2 \right) \right|^2}{\|\boldsymbol{Y}_{\widehat{\mathcal{T}},l}\|_2^2} \\
&\leq 2 \sum_{i \in \widehat{\mathcal{T}}} \frac{\left| Y_{il} - \beta u_i u_l \right|^2 + |u_i|^2 \left| \beta u_l - \|\boldsymbol{Y}_{\widehat{\mathcal{T}},l}\|_2 \right|^2}{\|\boldsymbol{Y}_{\widehat{\mathcal{T}},l}\|_2^2} \\
&\leq 2s \frac{\max_{i \in [d]} \left| Y_{il} - \mathbb{E}\left[ Y_{il} \right] \right|^2}{\|\boldsymbol{Y}_{\widehat{\mathcal{T}},l}\|_2^2} + 2 \frac{\left| \beta u_l - \|\boldsymbol{Y}_{\widehat{\mathcal{T}},l}\|_2 \right|^2}{\|\boldsymbol{Y}_{\widehat{\mathcal{T}},l}\|_2^2},
\end{aligned}
\tag{26}
$$

where we use $(a_1 + a_2)^2 \leq 2(a_1^2 + a_2^2)$ in the first inequality and use $\|\boldsymbol{u}_{\widehat{\mathcal{T}}}\|_2^2 \leq 1$ in the last inequality.

To estimate (26), we consider the following event:

$$\left\{ \max_{i \in \mathcal{T}^c} |Y_{il}| \leq \epsilon_5 = \frac{\varsigma_2 \zeta^2 \beta \|\boldsymbol{u}\|_\infty}{\sqrt{s}}, \max_{i \in \mathcal{T}} \left| Y_{il} - \mathbb{E}[Y_{il}] \right| \leq \epsilon_5, \mathcal{T}_\zeta^+ \subset \widehat{\mathcal{T}} \right\}, \tag{27}$$

where $\varsigma_2 > 0$ is a constant close to 0. This event is related to (16)(18)(19).

Under the event in (27), we first estimate the lower bound of $\|\boldsymbol{Y}_{\widehat{\mathcal{T}},l}\|_2^2$. It holds that

$$
\begin{aligned}
\|\boldsymbol{Y}_{\widehat{\mathcal{T}},l}\|_2^2 &\geq \sum_{i \in \mathcal{T}_\zeta^+} \left( \beta |u_i u_l| - |\beta u_i u_l - Y_{il}| \right)^2 \\
&\geq \sum_{i \in \mathcal{T}_\zeta^+} \left( \beta |u_i u_l| - \frac{\varsigma_2 \zeta^2 \beta \|\boldsymbol{u}\|_\infty}{\sqrt{s}} \right)^2 \\
&\geq \sum_{i \in \mathcal{T}_\zeta^+} \left( \beta |u_i u_l| - 4\varsigma_2 \beta |u_i u_l| \right)^2 \\
&\geq \frac{3}{4} (1 - 4\varsigma_2)^2 \beta^2 |u_l|^2,
\end{aligned}
\tag{28}
$$

where $\mathcal{T}_\zeta^+ \subset \widehat{\mathcal{T}}$ and triangle inequality are used in the first inequality, the third inequality holds by $l \in \mathcal{T}_2$ and $i \in \mathcal{T}_\zeta^+$ and the last inequality holds by $\|\boldsymbol{u}_{\mathcal{T}_\zeta^+}\|_2 \geq 1 - \frac{\zeta^2}{4} \geq \frac{3}{4}$.

Second, we estimate $\left| \beta u_l - \|\boldsymbol{Y}_{\widehat{\mathcal{T}},l}\|_2 \right|^2$. We obtain

$$
\begin{aligned}
\left| \beta u_l - \|\boldsymbol{Y}_{\widehat{\mathcal{T}},l}\|_2 \right|^2 &= \beta^2 \left| u_l \right|^2 - 2\beta u_l \|\boldsymbol{Y}_{\widehat{\mathcal{T}},l}\|_2 + \|\boldsymbol{Y}_{\widehat{\mathcal{T}},l}\|_2^2 \\
&\leq \beta^2 \left| u_l \right|^2 - 2\beta u_l \sqrt{\sum_{i \in \widehat{\mathcal{T}}} \left( \beta \left| u_i u_l \right| - \epsilon_5 \right)^2} + \sum_{i \in \widehat{\mathcal{T}}} \left( \beta \left| u_i u_l \right| + \epsilon_5 \right)^2 \\
&\leq \beta^2 \left| u_l \right|^2 - 2\beta u_l \sqrt{\beta^2 \left| u_l \right|^2 \|\boldsymbol{u}_{\widehat{\mathcal{T}}}\|_2^2 - 2\epsilon_5 \beta u_l \|\boldsymbol{u}_{\widehat{\mathcal{T}}}\|_1 + s\epsilon_5^2} \\
&\quad + \beta^2 \left| u_l \right|^2 \|\boldsymbol{u}_{\widehat{\mathcal{T}}}\|_2^2 + 2\epsilon_5 \beta u_l \|\boldsymbol{u}_{\widehat{\mathcal{T}}}\|_1 + s\epsilon_5^2,
\end{aligned}
$$

where the first inequality holds similar to (28). Thus we have

$$
\begin{aligned}
&\left| \beta u_l - \|\boldsymbol{Y}_{\widehat{\mathcal{T}},l}\|_2 \right|^2 \\
&\leq 2\beta u_l \left( \beta u_l \frac{\|\boldsymbol{u}_{\widehat{\mathcal{T}}}\|_2^2 + 1}{2} + \epsilon_5 \|\boldsymbol{u}_{\widehat{\mathcal{T}}}\|_1 - \sqrt{\beta^2 \left| u_l \right|^2 \|\boldsymbol{u}_{\widehat{\mathcal{T}}}\|_2^2 - 2\epsilon_5 \beta u_l \|\boldsymbol{u}_{\widehat{\mathcal{T}}}\|_1 + s\epsilon_5^2} \right) + s\epsilon_5^2,
\end{aligned} \tag{29}
$$

To complete this estimation, we compute

$$
\begin{aligned}
&\beta u_l \frac{\|\boldsymbol{u}_{\widehat{\mathcal{T}}}\|_2^2 + 1}{2} + \epsilon_5 \|\boldsymbol{u}_{\widehat{\mathcal{T}}}\|_1 - \sqrt{\beta^2 \left| u_l \right|^2 \|\boldsymbol{u}_{\widehat{\mathcal{T}}}\|_2^2 - 2\epsilon_5 \beta u_l \|\boldsymbol{u}_{\widehat{\mathcal{T}}}\|_1 + s\epsilon_5^2} \\
&= \frac{\left( \beta u_l \frac{\|\boldsymbol{u}_{\widehat{\mathcal{T}}}\|_2^2 + 1}{2} + \epsilon_5 \|\boldsymbol{u}_{\widehat{\mathcal{T}}}\|_1 \right)^2 - \left( \beta^2 \left| u_l \right|^2 \|\boldsymbol{u}_{\widehat{\mathcal{T}}}\|_2^2 - 2\epsilon_5 \beta u_l \|\boldsymbol{u}_{\widehat{\mathcal{T}}}\|_1 + s\epsilon_5^2 \right)}{\beta u_l + \epsilon_5 \|\boldsymbol{u}_{\widehat{\mathcal{T}}}\|_1 + \sqrt{\beta^2 \left| u_l \right|^2 \|\boldsymbol{u}_{\widehat{\mathcal{T}}}\|_2^2 - 2\epsilon_5 \beta u_l \|\boldsymbol{u}_{\widehat{\mathcal{T}}}\|_1 + s\epsilon_5^2}} \\
&\leq \frac{1}{\beta u_l} \left( \beta^2 \left| u_l \right|^2 \frac{\left( 1 - \|\boldsymbol{u}_{\widehat{\mathcal{T}}}\|_2^2 \right)^2}{4} + 4\epsilon_5 \beta u_l \|\boldsymbol{u}_{\widehat{\mathcal{T}}}\|_1 + \left( \|\boldsymbol{u}_{\widehat{\mathcal{T}}}\|_1^2 - s \right)\epsilon_5^2 \right), \\
&\leq \beta u_l \frac{\zeta^4}{64} + 4\epsilon_5 \sqrt{s},
\end{aligned}
$$

where we use $\|\boldsymbol{u}_{\widehat{\mathcal{T}}}\|_1 \leq \sqrt{s}$ and $1 - \frac{\zeta^2}{4} \leq \|\boldsymbol{u}_{\widehat{\mathcal{T}}}\|_2^2 \leq 1$ in the last inequality. It follows that (29) can be simplified as

$$
\left| \beta u_l - \|\boldsymbol{Y}_{\widehat{\mathcal{T}},l}\|_2 \right|^2 \leq \beta^2 \left| u_l \right|^2 \frac{\zeta^4}{32} + 8\epsilon_5 \beta u_l \sqrt{s} + s\epsilon_5^2. \tag{30}
$$

Therefore, under the event in (27), combining (24)(26)(28)(30), we have

$$
\begin{aligned}
&\mathrm{dist}\left( \frac{\boldsymbol{Y}_{\widehat{\mathcal{T}},l}}{\|\boldsymbol{Y}_{\widehat{\mathcal{T}},l}\|_2}, \boldsymbol{u}_{\widehat{\mathcal{T}}} \right)^2 \\
&\leq \frac{2s \frac{\varsigma_2^2 \zeta^4 \beta^2 \|\boldsymbol{u}\|_\infty^2}{s}}{\frac{3}{4}(1 - 4\varsigma_2)^2 \beta^2 \left| u_l \right|^2} + 2 \frac{\beta^2 \left| u_l \right|^2 \frac{\zeta^4}{32} + 8 \frac{\varsigma_2 \zeta^2 \beta \|\boldsymbol{u}\|_\infty}{\sqrt{s}} \beta u_l \sqrt{s} + s \frac{\varsigma_2^2 \zeta^4 \beta^2 \|\boldsymbol{u}\|_\infty^2}{s}}{\frac{3}{4}(1 - 4\varsigma_2)^2 \beta^2 \left| u_l \right|^2} \\
&\leq \left( \frac{32\varsigma_2^2}{3(1 - 4\varsigma_2)^2} + \frac{1}{12(1 - 4\varsigma_2)^2} + \frac{128\varsigma_2}{3(1 - 4\varsigma_2)^2} + \frac{32\varsigma_2^2}{3(1 - 4\varsigma_2)^2} \right) \zeta^2 \\
&\leq \frac{3}{4}\zeta^2,
\end{aligned} \tag{31}
$$

where in the second inequality we use $\left| u_l \right| \geq \frac{\|\boldsymbol{u}\|_\infty}{2}$ and $0 < \zeta \leq 1$, and in the last equality we set $\varsigma_2 = 1/69$. Then,

according to (31) and the event in (27), using (16)(18) with $\epsilon_3 = \epsilon_4 = \frac{\zeta^2 \beta \|\boldsymbol{u}\|_\infty}{69\sqrt{s}}$ and (19), it holds that

$$\mathbb{P}\left\{ \mathrm{dist}(\frac{\boldsymbol{Y}_{\widehat{\mathcal{T}},l}}{\|\boldsymbol{Y}_{\widehat{\mathcal{T}},l}\|_2}, \boldsymbol{u}_{\widehat{\mathcal{T}}}) \leq \frac{\sqrt{3}}{2}\zeta \right\} \geq 1 - \frac{69\sqrt{2s}(d-s)}{\sqrt{\pi}\zeta^2\beta\|\boldsymbol{u}\|_\infty} \exp\left( -\frac{\zeta^4\beta^2\|\boldsymbol{u}\|_\infty^2}{9522s} \right) - \frac{138\sqrt{s}s}{\sqrt{\pi}\zeta\beta\|\boldsymbol{u}\|_\infty} \exp\left( -\frac{\zeta^4\beta^2\|\boldsymbol{u}\|_\infty^2}{19044s} \right)$$
$$- \frac{8d}{3\sqrt{\pi}\beta\|\boldsymbol{u}\|_\infty^2} \exp\left( -\frac{9\beta^2\|\boldsymbol{u}\|_\infty^4}{256} \right) - \frac{8\sqrt{2s}s(d-s)}{\sqrt{\pi}\zeta\beta\|\boldsymbol{u}\|_\infty} \exp\left( -\frac{\zeta^2\beta^2\|\boldsymbol{u}\|_\infty^2}{128s} \right)$$
$$- \frac{16\sqrt{s}s^2}{\sqrt{\pi}\zeta\beta\|\boldsymbol{u}\|_\infty} \exp\left( -\frac{\zeta^2\beta^2\|\boldsymbol{u}\|_\infty^2}{256s} \right).$$

$$(32)$$

Finally, similar to (19), taking the union bound and using (15)(23)(31)(32), we obtain

$$\mathbb{P}\left\{ \mathrm{dist}(\hat{\boldsymbol{u}}_{\mathrm{nv}}, \boldsymbol{u}_{\widehat{\mathcal{T}}}) \leq \zeta \right\}$$
$$= \sum_{l \in \mathcal{T}_2} \mathbb{P}\left\{ \mathrm{dist}(\frac{\boldsymbol{Y}_{\widehat{\mathcal{T}},l}}{\|\boldsymbol{Y}_{\widehat{\mathcal{T}},l}\|_2}, \boldsymbol{u}_{\widehat{\mathcal{T}}}) \leq \frac{\sqrt{3}}{2}\zeta, i_0 = l \right\}$$
$$\geq \sum_{l \in \mathcal{T}_2} (1 - \mathbb{P}\left\{ \mathrm{dist}(\frac{\boldsymbol{Y}_{\widehat{\mathcal{T}},l}}{\|\boldsymbol{Y}_{\widehat{\mathcal{T}},l}\|_2}, \boldsymbol{u}_{\widehat{\mathcal{T}}}) > \frac{\sqrt{3}}{2}\zeta \right\} - \mathbb{P}\{i_0 \neq l\})$$
$$\geq \sum_{l \in \mathcal{T}_2} (\mathbb{P}\{i_0 = l\} - \mathbb{P}\left\{ \mathrm{dist}(\frac{\boldsymbol{Y}_{\widehat{\mathcal{T}},l}}{\|\boldsymbol{Y}_{\widehat{\mathcal{T}},l}\|_2}, \boldsymbol{u}_{\widehat{\mathcal{T}}}) > \frac{\sqrt{3}}{2}\zeta \right\})$$
$$\geq 1 - \frac{69\sqrt{2s}s(d-s)}{\sqrt{\pi}\zeta^2\beta\|\boldsymbol{u}\|_\infty} \exp\left( -\frac{\zeta^4\beta^2\|\boldsymbol{u}\|_\infty^2}{9522s} \right) - \frac{138\sqrt{s}s^2}{\sqrt{\pi}\zeta\beta\|\boldsymbol{u}\|_\infty} \exp\left( -\frac{\zeta^4\beta^2\|\boldsymbol{u}\|_\infty^2}{19044s} \right)$$
$$- \frac{8(1+s)d}{3\sqrt{\pi}\beta\|\boldsymbol{u}\|_\infty^2} \exp\left( -\frac{9\beta^2\|\boldsymbol{u}\|_\infty^4}{256} \right) - \frac{8\sqrt{2s}s^2(d-s)}{\sqrt{\pi}\zeta\beta\|\boldsymbol{u}\|_\infty} \exp\left( -\frac{\zeta^2\beta^2\|\boldsymbol{u}\|_\infty^2}{128s} \right)$$
$$- \frac{16\sqrt{s}s^3}{\sqrt{\pi}\zeta\beta\|\boldsymbol{u}\|_\infty} \exp\left( -\frac{\zeta^2\beta^2\|\boldsymbol{u}\|_\infty^2}{256s} \right).$$

If $\beta$ satisfies

$$\beta \geq 138\zeta^{-2}\|\boldsymbol{u}\|_\infty^{-1}\sqrt{s(\log d + 2\log s)}, \tag{33}$$

it implies that

$$\mathbb{P}\left\{ \mathrm{dist}(\hat{\boldsymbol{u}}_{\mathrm{nv}}, \boldsymbol{u}_{\widehat{\mathcal{T}}}) \leq \zeta \right\} \geq 1 - \frac{231\sqrt{2} + 470}{414\sqrt{\pi\log 2}} d^{-1} > 1 - 1.3041 d^{-1}.$$

$\square$

*Remark* A.8. From (33), a sufficient condition for the constant $C_2$ in Theorem 2.3 is

$$C_2 \geq 138\sqrt{3}.$$

### A.6. Proof of Theorem 2.4

*Proof.* Recall that $\mathcal{T}_\zeta^+ = \left\{ i \in \mathcal{T} : |u_i| \geq \frac{\zeta}{2\sqrt{s}} \right\}$. From the assumption of $\boldsymbol{u}$, we have $\mathcal{T} = \mathcal{T}_{2\theta}^+$. Therefore, using (19) and $\left|\widehat{\mathcal{T}}\right| = |\mathcal{T}| = s$, we obtain

$$\mathbb{P}\left\{ \widehat{\mathcal{T}} = \mathcal{T} \right\} \geq 1 - \frac{8d}{3\sqrt{\pi}\beta\|\boldsymbol{u}\|_\infty^2} \exp\left( -\frac{9\beta^2\|\boldsymbol{u}\|_\infty^4}{256} \right) - \frac{4\sqrt{2s}s(d-s)}{\sqrt{\pi}\theta\beta\|\boldsymbol{u}\|_\infty} \exp\left( -\frac{\theta^2\beta^2\|\boldsymbol{u}\|_\infty^2}{32s} \right)$$
$$- \frac{8\sqrt{s}s^2}{\sqrt{\pi}\theta\beta\|\boldsymbol{u}\|_\infty} \exp\left( -\frac{\theta^2\beta^2\|\boldsymbol{u}\|_\infty^2}{64s} \right).$$

If $\beta$ satisfies

$$\beta \geq \max\left\{ \frac{32\sqrt{2}}{3}\|\boldsymbol{u}\|_\infty^{-2}\sqrt{\log d}, 8\theta^{-1}\|\boldsymbol{u}\|_\infty^{-1}\sqrt{s(\log d + 2\log s)} \right\}, \tag{34}$$

it implies that

$$\mathbb{P}\left\{\widehat{\mathcal{T}} = \mathcal{T}\right\} \geq 1 - \frac{5 + 4\sqrt{2}}{4\sqrt{2\pi \log 2}} d^{-1} > 1 - 1.2766 d^{-1}.$$

$\square$

*Remark* A.9. From (34), a sufficient condition for the constant $C_3$ in Theorem 2.4 is

$$C_3 \geq \frac{32\sqrt{2}}{3}.$$

Theorem 2.4 for the spiked Wigner model can be extended to obtain the following support recovery guarantee for Algorithm 1 of Cai et al. (2025) under the single-spiked covariance model of Johnstone (2001).

**Theorem A.10.** *Let $v \in \mathbb{R}^p$ be a $k$-sparse unit vector satisfying $|v_i| \geq \vartheta/\sqrt{k}$ for all $i \in \mathcal{S}$ and some constant $\vartheta > 0$, where $\mathcal{S}$ denotes the support of $v$. Let $x_i = \sqrt{\lambda} g_i v + \xi_i$, $i = 1, \ldots, n$, where $\lambda > 0$, $g_i \overset{i.i.d.}{\sim} \mathcal{N}(0,1)$ and $\xi_i \overset{i.i.d.}{\sim} \mathcal{N}(\mathbf{0}, \mathbf{I}_p)$ are independent. There exists universal constants $C_6, C_7 > 0$ such that if*

$$\lambda \geq C_6 \|v\|_\infty^{-1} \quad and \quad n \geq C_7 \vartheta^{-2} k \log p,$$

*Algorithm 1 in Cai et al. (2025) recovers the support exactly (i.e., $\widehat{\mathcal{S}} = \mathcal{S}$ with the estimated support $\widehat{\mathcal{S}}$) with probability at least $1 - 3p^{-1}$.*

The proof of Theorem A.10 is similar to that of Theorem 2.4 and is therefore omitted.

### A.7. Proof of Theorem 3.1

The proof of Theorem 3.1 is organized into two parts. First, we show that $u^0$ falls into a small constant neighborhood of $u$. Subsequently, we prove the convergence of the truncated power method.

*Proof of Theorem 3.1.* We denote $\tilde{s} = s + 2k$, $\rho = \rho(W, \tilde{s})$ and $\mathcal{F}_t = \text{supp}(u^t)$, where $k = C_5 s$ for some absolute constant $C_5 \geq 1$. Similar to the proof of Theorem 3.1, setting $r = 0.01\zeta'$ and $\ell = \tilde{s}$ in Lemma A.3 with $\zeta' \in (0,1)$ and $\zeta = 1$ in Lemma A.6, if $\beta$ satisfies

$$\beta \geq \max\left\{16\|u\|_\infty^{-1}\sqrt{s(\log d + 2\log s)}, 200(\zeta')^{-1}\sqrt{(1 + 2C_5)s\log\left(\frac{9ed}{(1 + 2C_5)s}\right) + \log d}\right\}, \tag{35}$$

then with the probability exceeding

$$1 - \frac{1}{\sqrt{\pi \log(432e^3)}} d^{-1} - \frac{7 + 3\sqrt{2}}{6\sqrt{\pi \log 2}} d^{-1} > 1 - 1.4571 d^{-1},$$

the following event holds:

$$\left\{\rho \leq 0.03\zeta'\beta, \ \text{dist}(u^0, u) \leq 1\right\}.$$

We will continue the proof under this event.

*Step 1: Estimating $|u^\top u^0|$.* Since $1 \geq \text{dist}(u, u^0)^2 = 2 - 2|u^\top u^0|$, we have $|u^\top u^0| \geq 0.5$.

*Step 2: Convergence of truncated power method.* To prove (7), we will first show that $\text{dist}(u, u^t) \leq 1$ by induction.

We denote $\Lambda_t = \mathcal{F}_{t-1} \cup \mathcal{F}_t \cup \mathcal{T}$, then $|\Lambda_t| \leq s + 2k = \tilde{s}$. Also, we define

$$w^t = Y_{\Lambda_t} u^{t-1}/\|Y_{\Lambda_t} u^{t-1}\|_2, \tag{36}$$

hence we have $u^t = w_{\mathcal{F}_t}^t/\|w_{\mathcal{F}_t}^t\|_2$ and $\mathcal{F}_t$ is the set of indices with the $k$ largest absolute values in $w^t$. Let $\kappa$ be the ratio of the second largest (in absolute value) to the largest eigenvalue of $Y_{\Lambda_t}$. Then, since $\mathcal{T} \subset \Lambda_t$, similar to (11)(12), we obtain

$$\kappa = \frac{\max_{i \neq 1}|\lambda_i(Y_{\Lambda_t})|}{|\lambda_1(Y_{\Lambda_t})|} \leq \frac{\rho}{\beta\|u_{\Lambda_t}\|_2^2 - \rho} \leq \frac{0.03\beta}{\beta - 0.03\beta} = \frac{3}{97} < 1,$$

where in the second inequality we use $\rho \leq 0.03\zeta'\beta$ and $\zeta' < 1$.

Let $\bar{u}$ be a unit eigenvector corresponding to the largest eigenvalue of $Y_{\Lambda_t}$ and satisfying $u^\top \bar{u} \geq 0$. hence we have $\text{dist}(u, \bar{u}) = \|u - \bar{u}\|_2$. Then, using (36) and Lemma A.1, we have

$$\left|\bar{u}^\top w^t\right| \geq \left|\bar{u}^\top u^{t-1}\right| \left(1 + \frac{1}{2}(1 - \kappa^2)\left(1 - \left|\bar{u}^\top u^{t-1}\right|^2\right)\right),$$

which implies that

$$1 - \left|\bar{u}^\top w^t\right| \leq \left(1 - \left|\bar{u}^\top u^{t-1}\right|\right)\left(1 - \frac{1 - \kappa^2}{2}\left(\left|\bar{u}^\top u^{t-1}\right| + \left|\bar{u}^\top u^{t-1}\right|^2\right)\right). \tag{37}$$

Since $\mathcal{T} \subset \Lambda_t$, Lemma A.4 gives

$$\|u - \bar{u}\|_2^2 = \text{dist}(u, \bar{u})^2 \leq 2 - 2\frac{1}{\sqrt{1 + \frac{\rho^2}{(\beta - 2\rho)^2}}}$$
$$\leq \frac{\rho^2}{(\beta - 2\rho)^2} \leq \frac{(0.03\zeta'\beta)^2}{(\beta - 0.06\beta)^2} = \frac{9(\zeta')^2}{8836}, \tag{38}$$

where in the second inequality we use $1 - \frac{1}{\sqrt{1+a}} \leq \frac{a}{2}$ for $a \geq 0$, and in the last two inequalities we use $\rho \leq 0.03\zeta'\beta$ and $\zeta' < 1$. Note that the induction assumption $\text{dist}(u, u^{t-1}) \leq 1$ implies that $\left|u^\top u^{t-1}\right| \geq 0.5$, which with (38) further leads to

$$\left|\bar{u}^\top u^{t-1}\right| \geq \left|u^\top u^{t-1}\right| - \left|(u - \bar{u})^\top u^{t-1}\right|$$
$$\geq \left|u^\top u^{t-1}\right| - \|u - \bar{u}\|_2 \|u^{t-1}\|_2 \geq 0.5 - \frac{3}{94}. \tag{39}$$

Plugging (39) into (37), we have
$$1 - \left|\bar{u}^\top w^t\right| \leq 0.6568(1 - \left|\bar{u}^\top u^{t-1}\right|),$$

which is equivalent to
$$\text{dist}(\bar{u}, w^t) \leq 0.8105 \cdot \text{dist}(\bar{u}, u^{t-1}), \tag{40}$$

where we use $\|\bar{u}\|_2 = \|w^t\|_2 = \|u^{t-1}\|_2 = 1$. For unit vectors $\bar{u}, u^{t-1}, u$, we obtain

$$\text{dist}(\bar{u}, u^{t-1}) \leq \text{dist}(\bar{u}, u) + \text{dist}(u^{t-1}, u). \tag{41}$$

This is because

$$\text{dist}(\bar{u}, u) + \text{dist}(u^{t-1}, u) = \|\tau_1 \bar{u} - u\|_2 + \|u + \tau_2 u^{t-1}\|_2$$
$$\geq \|\tau_1 \bar{u} + \tau_2 u^{t-1}\|_2$$
$$\geq \text{dist}(\bar{u}, u^{t-1}),$$

where $\tau_1, \tau_2 \in \{\pm 1\}$ and we use (5). Similarly, for unit vectors $u, w^t, \bar{u}$, it holds that

$$\text{dist}(u, w^t) \leq \text{dist}(u, \bar{u}) + \text{dist}(w^t, \bar{u}). \tag{42}$$

Using (38)(40)(41)(42), we have

$$\text{dist}(u, w^t) \leq 0.8105 \cdot \text{dist}(u, u^{t-1}) + 0.0578\zeta'. \tag{43}$$

Since $k = C_5 s$ and $\mathcal{F}_t$ is the set of indices with the largest $k$ absolute values in $w^t$, Lemma A.2 generates

$$\left|u^\top w_{\mathcal{F}_t}^t\right| \geq \left|u^\top w^t\right| - C_5^{-1/2} \min\left\{\sqrt{1 - |u^\top w^t|^2}, (1 + C_5^{-1/2})\left(1 - |u^\top w^t|^2\right)\right\}$$
$$\geq \left|u^\top w^t\right| - C_5^{-1/2}(1 + C_5^{-1/2})\left(1 - |u^\top w^t|^2\right),$$

which implies that

$$1 - \left| \boldsymbol{u}^\top \boldsymbol{w}^t_{\mathcal{F}_t} \right| \leq 1 - \left| \boldsymbol{u}^\top \boldsymbol{w}^t \right| + C_5^{-1/2}(1 + C_5^{-1/2})\Big(1 - \left| \boldsymbol{u}^\top \boldsymbol{w}^t \right|^2 \Big) \leq D_1^2(1 - \left| \boldsymbol{u}^\top \boldsymbol{w}^t \right|),$$

where $D_1 := \sqrt{1 + 2C_5^{-1/2}(1 + C_5^{-1/2})}$. Then, since $\boldsymbol{u}^t = \boldsymbol{w}^t_{\mathcal{F}_t}/\|\boldsymbol{w}^t_{\mathcal{F}_t}\|_2$, we have

$$
\begin{aligned}
\operatorname{dist}(\boldsymbol{u}, \boldsymbol{u}^t) = \sqrt{2 - 2\left| \boldsymbol{u}^\top \boldsymbol{u}^t \right|} &= \sqrt{2 - 2\left| \boldsymbol{u}^\top \boldsymbol{w}^t_{\mathcal{F}_t} \right|/\|\boldsymbol{w}^t_{\mathcal{F}_t}\|_2} \\
&\leq \sqrt{2 - 2\left| \boldsymbol{u}^\top \boldsymbol{w}^t_{\mathcal{F}_t} \right|} \leq D_1 \cdot \sqrt{2(1 - \left| \boldsymbol{u}^\top \boldsymbol{w}^t \right|)} \\
&= D_1 \cdot \operatorname{dist}(\boldsymbol{u}, \boldsymbol{w}^t) \\
&\leq 0.8105 D_1 \cdot \operatorname{dist}(\boldsymbol{u}, \boldsymbol{u}^{t-1}) + 0.0578 D_1 \zeta'
\end{aligned}
\tag{44}
$$

where in the last second inequality we use (43). Since $\operatorname{dist}(\boldsymbol{u}, \boldsymbol{u}^{t-1}) \leq 1$ and $\zeta' < 1$, the above inequality also implies that $\operatorname{dist}(\boldsymbol{u}, \boldsymbol{u}^t) \leq 1$ with suitable constant $C_5$ (constant $D_1$). Therefore, we complete the induction, which proves that $\operatorname{dist}(\boldsymbol{u}, \boldsymbol{u}^t) \leq 1$ for all $t$. As a result, the above inequality holds for all $t$, which leads to

$$
\begin{aligned}
\operatorname{dist}(\boldsymbol{u}, \boldsymbol{u}^t) &\leq \eta \cdot \operatorname{dist}(\boldsymbol{u}, \boldsymbol{u}^{t-1}) + D_2 \zeta' \\
&\leq \eta^2 \cdot \operatorname{dist}(\boldsymbol{u}, \boldsymbol{u}^{t-2}) + \eta D_2 \zeta' + D_2 \zeta' \\
&\leq \cdots \\
&\leq \eta^t \cdot \operatorname{dist}(\boldsymbol{u}, \boldsymbol{u}^0) + h\zeta',
\end{aligned}
$$

where $\eta := 0.8105 D_1$, $D_2 := 0.0578 D_1$ and $h := \frac{D_2}{1-\eta}$. This inequality is just (7). $\qquad\square$

*Remark* A.11. From (35)(44), a sufficient condition for constants $C_4, C_5$ in Theorem 3.1 is

$$C_4 \geq \max\left\{ 16\sqrt{3}, 200\sqrt{1 + (1 + 2C_5)(1 + \log_2(9\mathrm{e}))} \right\},$$

$$(0.8105 + 0.0578)\sqrt{1 + 2C_5^{-1/2}(1 + C_5^{-1/2})} < 1.$$

It can be simplified as

$$C_4 \geq 200\sqrt{(1 + 2\log_2(9\mathrm{e}))C_5 + 2 + 2\log_2(9\mathrm{e})},$$
$$C_5 \geq 49.047.$$

## B. Additional Experimental Results

### B.1. Computational Efficiency and Statistical Performance

Figure 7 provides complementary analysis of computational efficiency and support recovery, extending results in Figures 5. Runtime results (left panel) show that our TPM matches the efficiency of diagonal thresholding (DT), with only modest increase in cost at small $\beta$ due to extra iterations with weaker initialization; in contrast, covariance thresholding (CT) and spectral projection (SP) are substantially more expensive across all signal strengths. Dimension-scaling results (middle and right panels) demonstrate that our TPM achieves high estimation accuracy and perfect support recovery (success rate = 1) across all tested dimensions, while competing methods underperform; in particular, DT maintains a high estimation error and low success rate throughout. This superior statistical performance incurs minimal computational overhead, as TPM's runtime scales comparably to the efficient DT baseline (see Figure 5).

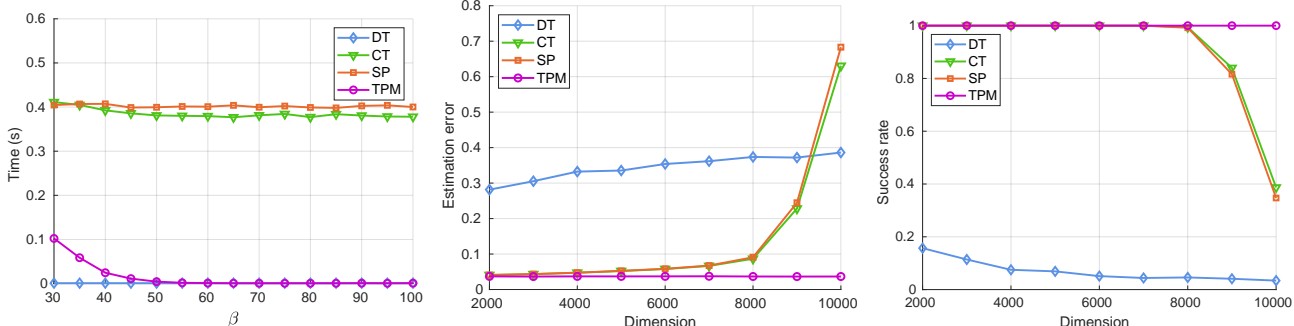

*Figure 7.* Left—Runtime versus signal strength $\beta$. Middle—Estimation error versus dimension $d$. Right—Support-recovery success rate versus dimension $d$. Experimental settings match those in Figures 5.

### B.2. Performance Trade-offs Between Column Thresholding and Its Variant

We evaluate the performance trade-off between column thresholding (Algorithm 1) and its computationally efficient variant (Algorithm 2). The two methods differ only in their estimation procedures: column thresholding applies eigenvalue decomposition to selected submatrices for higher accuracy, whereas the variant adopts a simpler normalization-based estimator for speed. Both use the same thresholding rule for support identification, yielding identical support recovery. We assess this trade-off along two axes: problem dimension and sparsity level.

Table 1 shows performance across increasing dimensions. Column thresholding consistently achieves substantially lower estimation error than the variant, but requires more computation time. These findings suggest that practitioners should choose column thresholding when estimation accuracy is paramount and the variant when computational budget is tight. In addition, the runtime of column thresholding grows almost linearly with dimension, reflecting favorable scaling relative to CT and SP that require $O(d^3)$ operations.

*Table 1.* Performance comparison of column thresholding (Algorithm 1) and its variant (Algorithm 2) across increasing dimensions. Column thresholding achieves lower estimation error at higher computational cost, while the variant offers faster runtimes with reduced estimation accuracy; support-recovery success rates are identical due to the shared thresholding strategy. Results averaged over 500 trials with $s = 25$ and $\beta = 150$.

| Algorithm | Metric | $d = 1000$ | $d = 3000$ | $d = 5000$ | $d = 7000$ | $d = 9000$ |
|---|---|---|---|---|---|---|
| | Estimation error | 0.0555 | 0.0846 | 0.0950 | 0.0993 | 0.1152 |
| Algorithm 1 | Success rate | 0.8660 | 0.7080 | 0.6480 | 0.6300 | 0.5420 |
| | Runtime (s) | $1.3\times10^{-3}$ | $1.4\times10^{-3}$ | $1.5\times10^{-3}$ | $1.6\times10^{-3}$ | $1.7\times10^{-3}$ |
| | Estimation error | 0.1932 | 0.2087 | 0.2165 | 0.2205 | 0.2305 |
| Algorithm 2 | Success rate | 0.8660 | 0.7080 | 0.6480 | 0.6300 | 0.5420 |
| | Runtime (s) | $1.9\times10^{-4}$ | $2.8\times10^{-4}$ | $3.8\times10^{-4}$ | $5.5\times10^{-4}$ | $7.2\times10^{-4}$ |

Table 2 examines performance under varying sparsity. Column thresholding preserves its accuracy advantage across all sparsity levels while incurring higher runtime. As sparsity increases, its runtime grows because the eigenvalue decompositions

operate on larger $s \times s$ submatrices. By contrast, the variant's runtime remains nearly constant across sparsity levels, being driven primarily by the ambient dimension rather than sparsity.

*Table 2.* Performance comparison of column thresholding (Algorithm 1) and its variant (Algorithm 2) under varying sparsity levels. Column thresholding maintains superior estimation accuracy at higher computational cost, while both methods achieve identical support recovery. Results averaged over 500 trials with $d = 5000$ and $\beta = 150$.

| Algorithm | Metric | $s = 10$ | $s = 15$ | $s = 20$ | $s = 25$ | $s = 30$ |
|---|---|---|---|---|---|---|
| Algorithm 1 | Estimation error | 0.0198 | 0.0243 | 0.0307 | 0.1030 | 0.2549 |
| | Success rate | 1 | 1 | 0.9900 | 0.6000 | 0.0620 |
| | Runtime (s) | $1.0 \times 10^{-3}$ | $1.1 \times 10^{-3}$ | $1.1 \times 10^{-3}$ | $1.5 \times 10^{-3}$ | $1.5 \times 10^{-3}$ |
| Algorithm 2 | Estimation error | 0.0738 | 0.1097 | 0.1456 | 0.2237 | 0.3619 |
| | Success rate | 1 | 1 | 0.9900 | 0.6000 | 0.0620 |
| | Runtime (s) | $4.1 \times 10^{-4}$ | $4.0 \times 10^{-4}$ | $4.0 \times 10^{-4}$ | $4.1 \times 10^{-4}$ | $4.1 \times 10^{-4}$ |

### B.3. Performance under Unknown Sparsities

Although the main presentation uses the true sparsity level $s$ for clarity, our method naturally admits a simple data-driven variant based on an elbow procedure. According to Section 2.1, we first choose

$$i_0 = \arg\max_i Y_{ii},$$

which is intended to select an index in the support. Similar to (4), when $i_0 \in \mathcal{T}$, the corresponding column satisfies

$$\mathbb{E}\big[Y\big]_{i,i_0} = \begin{cases} \beta\, u_i\, u_{i_0}, & i \in \mathcal{T}, \\ 0, & i \in \mathcal{T}^c. \end{cases}$$

After sorting the magnitudes of this column in decreasing order, say $|a_{(1)}| \geq |a_{(2)}| \geq \cdots \geq |a_{(d)}|$, the entries corresponding to indices in $\mathcal{T}$ are typically separated from those in $\mathcal{T}^c$, whose means are zero. We estimate $s$ by an elbow procedure: we compute the consecutive ratios $|a_{(k)}|/(|a_{(k+1)}| + \epsilon)$ of the sorted magnitudes and select the position of the largest ratio, where $\epsilon > 0$ is a small constant added to avoid division by zero. Because the out-of-support entries are noise-dominated, this ratio typically peaks at the support boundary, thereby locating the elbow automatically. The recovery procedure then runs with $\hat{s}$ in place of $s$. This extended version has essentially the same computational cost as the original algorithm.

Table 3 shows that in the moderate-to-high signal regime, the resulting performance is very close to that of the oracle version. In our experiments with dimension $d = 1000$, sparsity $s = 10$, and 200 trials, for $\beta \geq 80$ the estimated sparsity is essentially exact, and the downstream estimation error and success rate are nearly identical to those obtained with the true sparsity.

*Table 3.* Comparison between the oracle version (using the true sparsity $s$) and the estimated-sparsity version (using the elbow estimate $\hat{s}$), for $d = 1000$, $s = 10$, and 200 trials.

| $\beta$ | 20 | 40 | 60 | 80 | 100 | 120 |
|---|---|---|---|---|---|---|
| Mean of $\hat{s}$ | 1.120 | 2.900 | 8.885 | 10.000 | 10.000 | 10.000 |
| Standard deviation of $\hat{s}$ | 0.408 | 2.526 | 2.733 | 0.174 | 0.000 | 0.000 |
| Estimation error (true $s$) | 1.1789 | 0.2019 | 0.0553 | 0.0376 | 0.0292 | 0.0246 |
| Estimation error (estimated $\hat{s}$) | 1.3504 | 0.9965 | 0.2230 | 0.0422 | 0.0292 | 0.0246 |
| Success rate (true $s$) | 0.160 | 0.905 | 0.995 | 1.000 | 1.000 | 1.000 |
| Success rate (estimated $\hat{s}$) | 0.000 | 0.005 | 0.665 | 0.970 | 1.000 | 1.000 |

### B.4. Empirical Signal Strength Requirements under Uniform Amplitudes

We examine the empirical signal strength requirement of the TPM initialized by column thresholding (Algorithm 3) in the *uniform-amplitude* setting. Our theoretical results show that, under the non-uniform $\ell_\infty$ condition $\|u\|_\infty = \Omega(1)$, a signal strength of order $\beta = \Omega\big(\sqrt{s \log d}\big)$ suffices. By contrast, the experiments here indicate that with uniform amplitudes, where $\|u\|_\infty = 1/\sqrt{s}$, the algorithm empirically requires a stronger signal of order $\beta = \Omega\big(s^{0.6}\sqrt{\log d}\big)$.

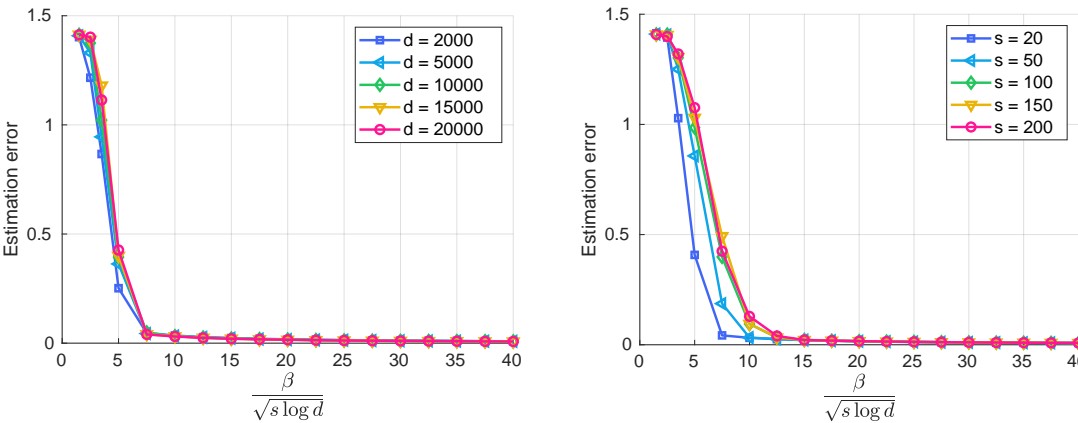

*Figure 8.* Estimation error versus scaled signal strength for TPM initialized by column thresholding (Algorithm 3), under varying dimensions (left) and sparsities (right). Experimental settings match those in Figure 2, except that the nonzero entries of the true spike $\boldsymbol{u}$ have uniform amplitudes.

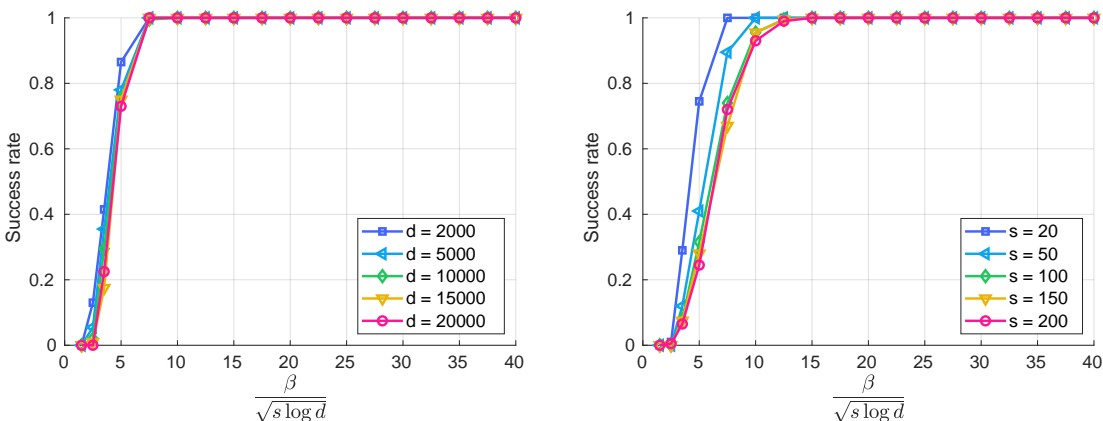

*Figure 9.* Success rate versus scaled signal strength for TPM initialized by column thresholding (Algorithm 3), under varying dimensions (left) and sparsities (right). Experimental settings match those in Figure 3, except that the nonzero entries of the true spike $\boldsymbol{u}$ have uniform amplitudes.

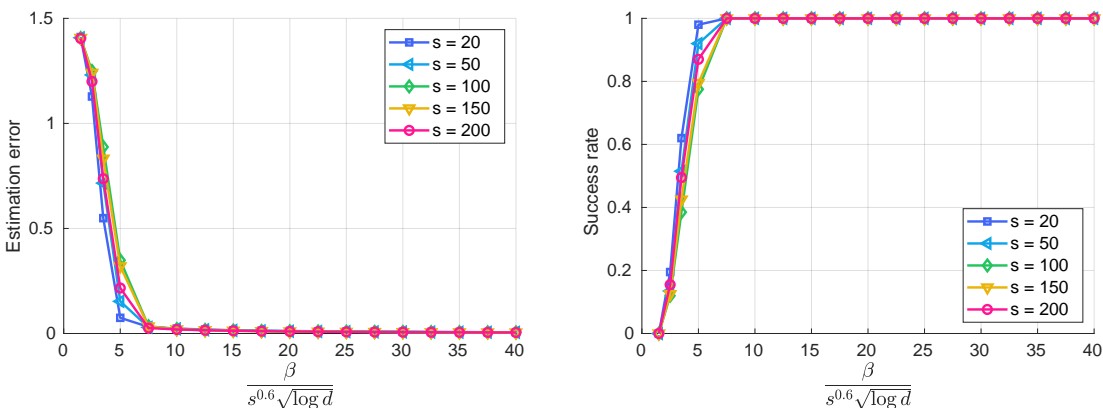

*Figure 10.* Estimation error (left) and success rate (right) versus scaled signal strength $\frac{\beta}{s^{0.6}\sqrt{\log d}}$ for the truncated power method initialized by column thresholding (Algorithm 3) under varying dimensions. Experimental settings match those in Figures 8 and 9.

Figure 8 reports the estimation error as a function of the scaled signal strength $\beta/\sqrt{s \log d}$. In panel (a), when varying the dimension $d$, the curves collapse under this scaling. In panel (b), however, when varying the sparsity level $s$, the curves stabilize at different phase-transition thresholds, with larger $s$ requiring larger values of $\beta/\sqrt{s \log d}$.

Figure 9 shows the corresponding support recovery performance. The phase-transition curves exhibit the same dependence on $s$: larger sparsity levels demand larger $\beta/\sqrt{s \log d}$ at transition. This parallel behavior indicates that the $\sqrt{s \log d}$ scaling is not attained for either estimation error or support identification in the uniform-amplitude case.

In Figure 10, we instead scale the signal strength by $\beta/(s^{0.6}\sqrt{\log d})$. Under this scaling, the curves for different $s$ align much more closely for both estimation error and support recovery. This collapse provides empirical evidence for an $s^{0.6}\sqrt{\log d}$ signal strength requirement for the column-thresholding-initialized truncated power method when the spike has uniform amplitudes.

Taken together, these experiments suggest that the $\ell_\infty$ condition in our analysis is not merely technical, but crucial for achieving the $\sqrt{s \log d}$ signal strength rate. In the uniform-amplitude setting, the algorithm does not empirically attain the $\sqrt{s \log d}$ scaling, in line with our theory. Nonetheless, the method remains practically attractive even when the $\ell_\infty$ condition is violated.

