# OpenReview forum: "Column Thresholding for Sparse Spiked Wigner Models: Improved Signal Strength Requirements"
_ICML.cc/2026/Conference — ICML 2026 regular_

### Official Review · Reviewer_NbmX · 2026-03-12

**Soundness:** 3
**Presentation:** 3
**Significance:** 3
**Originality:** 3
**Overall Recommendation:** 5
**Confidence:** 3

**Summary:**

This manuscript focuses on the computational-statistical gap in the sparse spiked Wigner model, where the information-theoretic optimal signal strength is $\tilde{\Omega}(\sqrt{s})$ but polynomial-time algorithms can only reach $\tilde{\Omega}(s)$. The authors propose a column thresholding approach that achieves the optimal $\tilde{\Omega}(\sqrt{s})$ scaling under the $\||u\||_\infty=\Omega(1)$ non-uniform spike condition, and design a truncated power method with linear convergence using column thresholding as initialization. Rigorous theoretical proofs and comprehensive experiments verify the effectiveness, efficiency and theoretical scaling of the proposed methods.

**Compliance With Llm Reviewing Policy:**

Affirmed.

**Final Justification:**

My main concerns have largely been addressed, and I will keep my current score. Overall, the authors’ response is sufficient and has alleviated most of my previous concerns.

**Key Questions For Authors:**

1. The assumption $||u||_\infty=\Omega(1)$ seems restrictive. Could the authors discuss possible relaxations of this condition, or analyze the performance of the method in the near-uniform sparse regime?

2. The manuscript claims that the proposed method avoids the planted-clique hardness barrier, but this point currently lacks a formal theoretical justification. Could the authors provide a more rigorous separation argument or a reduction-based explanation?

3. Please clarify why existing initialization methods cannot achieve the optimal $\widetilde{\Omega}(\sqrt{s})$ scaling in this setting, and more explicitly highlight the core novelty of the proposed initialization relative to prior work.

**Limitations:**

yes

**Strengths And Weaknesses:**

Strengths

1. Soundness: The theoretical derivation is complete and rigorous, all core theorems have complete proofs, and the experimental results are highly consistent with the theoretical phase-transition scaling, ensuring the technical correctness of the work.

2. Originality: The column thresholding strategy breaks through the traditional diagonal/global thresholding framework, providing a new way to close the computational-statistical gap for non-uniform sparse spikes.

3. Significance: It solves the recovery problem of a typical class of non-uniform spikes in the sparse spiked Wigner model, and provides a high-quality initialization scheme for non-convex sparse PCA, which has reference value for related high-dimensional statistical inference tasks.

4. Presentation: The paper structure is clear, the experimental design is systematic, and the comparison with baseline methods is sufficient.

Weaknesses

1. Soundness: The $\||u\||_\infty=\Omega(1)$ condition is excessively restrictive, lacking relaxation analysis and robustness verification for signals that do not fully satisfy this constraint.

2. Originality: The truncated power method is a classic existing algorithm, and the innovation of the paper only lies in the initialization module, which is incremental rather than subversive.

3. Significance: The method is invalid for uniform spikes, and does not solve the core open problem of the field, so the application scope is greatly limited.

4. Presentation: The proof appendix is too dense, lacking intuitive proof sketches, and the explanation of algorithm variants is not friendly to practical users.

---

> ### Author Rebuttal · Authors · 2026-03-31
>
> >Q1. The condition $||\pmb u||\_\infty=\Omega(1)$ seems restrictive.
>
> **Reply:** We thank the reviewer for this comment. **The condition $||\pmb u||\_\infty=\Omega(1)$ is not a hard requirement of our method**; it is the regime where our guarantee is strongest. Theorem 2.2 applies throughout the admissible range $||\pmb u||\_\infty\in[s^{-1/2},1]$ and gives the required signal strength $\widetilde\Omega(\sqrt{s}||\pmb{u}||\_\infty^{-1})$, so the guarantee degrades smoothly---not abruptly---as the spike becomes more uniform: **from $\widetilde\Omega(\sqrt s)$ when $||\pmb u||\_\infty=\Omega(1)$, through $\widetilde\Omega(s^{1/2+\alpha})$ when $||\pmb u||\_\infty\asymp s^{-\alpha}$, to $\widetilde\Omega(s)$ for the uniform spike**.
>
> Thus, **our method is robust to partial violations of the $\ell_\infty$ condition**: the guarantee degrades smoothly as the spike becomes more uniform, rather than failing abruptly outside the $\Omega(1)$ regime. We will revise the paper to clarify that the $\ell_\infty$ condition identifies the most favorable regime, not a sharp boundary of applicability. Whether the $||\pmb u||\_\infty^{-1}$ dependence is optimal remains an interesting open question.
>
> >Q2. The method avoids planted-clique hardness barrier.
>
> **Reply:** We thank the reviewer for this comment and agree that our wording should be more precise. **Our result is a *structural separation*, not an algorithm that overcomes planted-clique hardness for the canonical sparse spiked Wigner model.**
>
> **The planted-clique reduction in Brennan et al. (2018) produces a *homogeneous* spike**: the planted-clique model is permutation-symmetric on the support $\mathcal{S}$, so the induced signal is proportional to $\mathbf{1}\_\mathcal{S}\mathbf{1}\_\mathcal{S}^\top$ with $||\pmb u||\_\infty=s^{-1/2}$.
>
> **Our improvement relies on *non-uniformity***: the initializer exploits column-level heterogeneity created by larger spike coordinates. When this heterogeneity is absent, the advantage disappears---specializing Theorem 2.2 to the homogeneous spike recovers the threshold $\widetilde\Omega(s)$, consistent with the known easy regime and the planted-clique conjecture.
>
> In other words, the Brennan et al. reduction does not establish hardness for the non-uniform regime where our theorem improves the threshold; our algorithm improves on a different spike class, not on planted-clique-derived instances. We will revise the paper to make this distinction explicit.
>
> >Q3. Clarify why existing initialization method cannot achieve the optimal scaling in this setting, and why our contribution on initialization is not incremental.
>
> **Reply:** We thank the reviewer for this comment. In sparse spiked Wigner model, truncated power iteration is a standard refinement whose convergence follows from existing results (e.g., Yuan and Zhang (2013)) once the initializer has nontrivial correlation with the spike. **The achievable signal-strength regime is therefore determined by the initialization, which is the main bottleneck.**
>
> **Non-uniformity can lower this bottleneck, but only when the initialization is based on a statistic that can detect it.** Diagonal thresholding relies on diagonal statistics whose gap $g_{\text{diag}}$ does not improve under $\ell_\infty$-type non-uniformity, so it inherently requires signal strength $\widetilde\Omega(s)$. **Our column thresholding is designed to exploit this structure**: by aggregating off-diagonal information at the column level, where larger spike coordinates create a detectable signal, it achieves a strictly larger gap $g_{\text{col}}\gg g_{\text{diag}}$ (Section 2.1), reducing the required signal strength to $\widetilde\Omega(\sqrt s)$ when $||\pmb u||\_\infty=\Omega(1)$. **To our knowledge, no prior polynomial-time initialization achieves this scaling.** We will revise the paper to make this distinction explicit.
>
> >Q4. The method does not apply to uniform spikes and does not resolve the core open problem, which greatly limits its scope.
>
> **Reply:** We thank the reviewer for this comment. **Our paper is not intended to resolve the uniform-spike case**, since under the planted clique conjecture, the impossible/hard/easy regime for uniform spikes in the spiked Wigner model is already understood (see our Figure 1). The corresponding statistical–computational gap is believed to be fundamental. Accordingly, one should not expect a polynomial-time method to close this gap in the uniform setting.
>
> **Our focus is instead on the non-uniform spike case, whose computational boundary remains largely unknown.** The main contribution of the paper is to identify a natural class of non-uniform spikes, characterized by an $\ell_\infty$ condition, for which we give a polynomial-time recovery method at signal strength $\widetilde\Omega(\sqrt s)$. This matches the information-theoretic limit of the uniform case and shows that the computational barrier for uniform spikes does not extend to all sparse spikes.

---

> > ### Author Rebuttal · Reviewer_NbmX · 2026-04-02
> >
> > I have read the authors' rebuttal and the other reviews. I thank the authors for their detailed responses, which have largely addressed my previous questions. I maintain my 'Accept' score and support the publication of this paper.

---

> > > ### Author Response · Authors · 2026-04-02
> > >
> > > Thank you for your follow-up review. We appreciate your time, positive assessment, and support for our paper.

---

### Official Review · Reviewer_jvJ8 · 2026-03-12

**Soundness:** 1
**Presentation:** 3
**Significance:** 2
**Originality:** 2
**Overall Recommendation:** 4
**Confidence:** 3

**Summary:**

The authors study the sparse spiked Wigner model, where an unknown sparse vector $u \in \mathbb{R}^d$ is observed through the noisy matrix $Y = \beta u u^\top + W$, with $W \sim \mathrm{GOE}(d)$. As a function of the sparsity $s$, different related works have characterized the presence of a statistical-to-computational gap, i.e., a difference between the Information Theoretic threshold (IT) $O(\sqrt{s})$ and the performance of polynomial-time algorithms $O(s)$. These results leverage a reduction to the planted clique problem. However, the authors stress that these results are based on a uniformity assumption of the vector $u$. Therefore, they claim the interest of studying the non-unfiorm case by closing the statistical-to-computational gap with a novel proposed algorithm based on column thresholding.

**Compliance With Llm Reviewing Policy:**

Affirmed.

**Final Justification:**

I have modified my score to the positive side, following the clear explanations of the authors during the rebuttal.

**Key Questions For Authors:**

- Can the author clarify the relation with Yuan & Zhang 2013? It is just briefly mentioned, but it seems a relevant point.
- Can the author comment on what the expected results are if one does not know the sparsity level? All their algorithms use the knowledge of the sparsity level, but could one use an elbow procedure for different sparsity levels? Given the varied audience of ICML, connections to more "practical" scenarios are relevant. Relatedly, expanding the connections with the rectangular case (clustering, etc.) would be interesting for the typical reader.
- Can the author comment on the phase transition present in this case? It would be nice to link the discontinuous behaviour with the presence of an algorithmic hard phase (first-order transitions, 0/1 transitions, etc.)

**Limitations:**

I do not see potential negative societal implications due to the theoretical nature of this work.

**Strengths And Weaknesses:**

The theoretical analysis is sound, and the presentation is rigorous at the mathematical level, with assumptions, results, and proofs detailed correctly.


The main concerns for this submission are what one could characterize as "significance" and "originality". However, these buckets do not capture the main problem in my opinion: there is an extensive literature based on the planted clique conjecture. This conjecture is only cited briefly, never explained in detail, and the reader must understand that it is concerned with the uniform case. How can a reader be convinced about the submission's relevance if the authors do not analyze why the planted clique does not cover the non-uniform case? Why is it not trivial to consider the case where a component is extremely larger than the other? This comment is meant to criticize the lack of a clear explanation of why one should be interested in the non-uniform case and not the fact that it is not interesting.


There is no mention of Approximate Message Passing, Overlap Gap property, and other notions that are extremely relevant in theoretical computer science to study algorithmic phase transitions. Optimality claims for polynomial-time algorithms exploit these tools.


The numerical experiments are carried out in numerical ranges that make the plots feel flat and missing the transition, although I acknowledge that this is a general problem of sharp transitions. The algorithm proposed has roots in a 2013 paper, and the comments on this are rather brief.

---

> ### Author Rebuttal · Authors · 2026-03-31
>
> >Q1. Clarify relation with Yuan and Zhang (2013)
>
> **Reply:** We thank the reviewer for this question. Yuan and Zhang (2013) focus on the iterations itself: they analyze the convergence of truncated power iteration and show that it recovers a sparse leading eigenvector at near-optimal rates, assuming a suitable initialization exists. Our paper addresses a different question. We study sparse spiked Wigner model and aim to characterize the signal-strength threshold at which polynomial-time estimation becomes possible. The main contribution is our column-thresholding and its analysis, which are what determine this threshold. **Truncated power iteration serves only as a standard refinement step after our initialization; it does not sharpen the theoretical guarantee but improves finite-sample performance.** We will clarify this in the revision.
>
> >Q2. Comment on what expected results are if one does not know sparsity
>
> **Reply:** We thank the reviewer for this comment. Although the main presentation uses true sparsity $s$ for clarity, the method admits a simple data-driven variant. We first select $i_0=\arg\max_i|Y_{ii}|$, which targets an index in the support. When $i_0\in\mathcal{T}$, the entries of column $\pmb{Y}\_{:,i_0}$ have nonzero mean on $\mathcal{T}$ and zero mean off $\mathcal{T}$, so after sorting their magnitudes in decreasing order, $|a\_{(1)}|\ge\cdots\ge|a\_{(d)}|$, a gap appears at the support boundary. We locate this gap by computing consecutive ratios $|a\_{(k)}|/(|a\_{(k+1)}|+\epsilon)$ (with a small $\epsilon>0$ to avoid division by zero): because out-of-support entries are noise-dominated, this ratio peaks sharply at the boundary, giving an automatic estimate $\hat s$. The recovery procedure then runs with $\hat s$ in place of $s$ at essentially the same computational cost.
>
> **Table 1:Comparison between oracle version (true $s$) and estimated-sparsity version (elbow estimate $\hat s$) for $d=1000,s=10$, $200$ trials.**
> |$\beta$|40|60|80|100|120|
> |-|-|-|-|-|-|
> |Mean of $\hat s$|2.9|8.9|10|10|10|
> |SD of $\hat s$|2.526|2.733|0.174|0|0|
> |Estimation error ($s$)|0.202|0.055|0.038|0.029|0.025|
> |Estimation error ($\hat s$)|0.997|0.223|0.042|0.029|0.025|
> |Success rate ($s$)|0.905|0.995|1|1|1|
> |Success rate ($\hat s$)|0.005|0.665|0.970|1|1|
>
> **Table 1 shows that performance of two versions are very close in moderate-to-high signal regime.** For $\beta\ge80$, estimated $\hat s$ is essentially exact, and the estimation error and success rate are nearly identical to those obtained with true $s$.
>
> We will also discuss the rectangular setting $\pmb{Y}\approx\beta\pmb u\pmb{v}^\top+\pmb W$, where the diagonal is unavailable but informative rows and columns can be identified via their norms or leading singular vectors, connecting to biclustering. Both additions will appear in the revision.
>
> >Q3. Comment on phase transition
>
> **Reply:** We thank the reviewer for this comment. Figures 2 and 3 indeed suggest a sharp phase transition in $\beta/\sqrt{s\log d}$: the estimation error drops abruptly, the support-recovery probability jumps from near 0 to 1 over a narrow range, and the curves collapse across different $(d,s)$ pairs, consistent with the scaling in Theorems 2.2 and 2.4. However, our results establish a computational threshold for the column-thresholding method, not the information-theoretic limit, so **the observed sharpness should be interpreted as an algorithmic transition rather than as evidence of a distinct hard phase.** We will clarify this in the revision.
>
> >Q4. No mention of AMP and OGP
>
> **Reply:** We thank the reviewer for this comment and agree that the manuscript should better acknowledge the broader literature on computational barriers. AMP is a powerful framework whose performance can be tracked via state evolution, yielding sharp threshold predictions and, in some cases, Bayes-optimal performance in spiked settings [1-2]. OGP plays a complementary role: it is a structural property--a fragmentation of the near-optimal solution space--widely used to formalize algorithmic hardness in random optimization and inference [3-6]. We will revise the introduction to discuss both and incorporate the relevant references.
>
> References:
> [1] Estimation of low-rank matrices via approximate message passing, Ann. Stat., 2021.
> [2] Constrained low-rank matrix estimation: Phase transitions, approximate message passing and applications, J. Stat. Mech.: Theory Exp., 2017.
> [3] The overlap gap property: A topological barrier to optimizing over random structures, PNAS, 2021.
> [4] The overlap gap property and approximate message passing algorithms for p-spin models, Ann. Probab., 2021.
> [5] Free energy wells and overlap gap property in sparse PCA, COLT, 2020.
> [6] The overlap gap property in principal submatrix recovery, Probab. Theory Relat. Fields., 2021.
>
> >Q5. Lack of explanation of why one should be interested in non-uniform case
>
> **Reply:** Thanks. Please refer to the response of Q2 for Reviewer ubzF.

---

> > ### Author Rebuttal · Reviewer_jvJ8 · 2026-04-02
> >
> > I sincerely thank the authors for their detailed rebuttal that addressed my concerns.

---

> > > ### Author Response · Authors · 2026-04-03
> > >
> > > Thank you for the positive acknowledgement and raising the score. We appreciate your time and thoughtful consideration.

---

### Official Review · Reviewer_ubzF · 2026-03-25

**Soundness:** 4
**Presentation:** 4
**Significance:** 3
**Originality:** 3
**Overall Recommendation:** 5
**Confidence:** 3

**Summary:**

The authors study the spiked Wigner model in a setting where the spike is d-dimensional and s-sparse, and focus in particular on the statistical to computational gap in signal strength (IT requires Omega(sqrt(s)), but common algorithms require Omega(s) SNR).
Given that the gap is conjectured fundamental for uniform spikes, the authors set forth to study if this gap persists for other classes of spikes, crucially excluding the uniform one from their analysis.

They prove that under a condition on the $\ell_{\infty}$ norm of the spike, column thresholding recovers the signal at SNR Omega(sqrt(s)), closing the gap. They then use column-thresholding as the warm initialisation of a two-stage algorithm to refine initial estimates.

The authors then validate their results numerically, doing finite-size scaling analysis and comparing performance and scaling time against other common algorithms, showing superiority of column thresholding.

**Compliance With Llm Reviewing Policy:**

Affirmed.

**Final Justification:**

As mentioned in the original review, I did not raise specific concerns with the paper. Results are interesting and seem sound, hence my recommendation is to accept the paper.

**Key Questions For Authors:**

I do not have specific questions

Minor points:
- s-sparsity seems to be never defined. While standard, maybe for a broader audience it would be worth reminding its meaning, for example in the Notation paragraph.

**Limitations:**

yes

**Strengths And Weaknesses:**

Strengths:
- the paper is written very clearly, and I appreciated in particular the review of previous results in the introduction and the intuitive justification for the gain using column thresholding, making it accessible to broader audiences
- the result is interesting: assessing the robustness of stat-to-comp gaps under changes in hidden-signal structure is a very nice idea
- the experiments are solid, confirm the theory, and provide accurate finite-size scaling analysis
- the theory seems solid

Note: I did not check proofs or the provided code

Weaknesses:
- I did not identify specific weaknesses

---

> ### Author Rebuttal · Authors · 2026-03-31
>
> >Q1. $s$-sparsity seems to be never defined. While standard, maybe for a broader audience it would be worth reminding its meaning, for example in the Notation paragraph.
>
> **Reply:** We thank the reviewer for this helpful suggestion. We agree that explicitly defining $s$-sparsity would improve readability, especially for readers outside the sparse estimation literature.
>
> In the revised manuscript, we will add the following definition in the Notation paragraph. A vector $\pmb u\in\mathbb{R}^n$ is called *$s$-sparse* if it has at most $s$ nonzero entries, i.e., $||\pmb u||\_0\le s$, where$$||\pmb u||\_0:=\bigl|\lbrace i\in[n]:u_i\neq0\rbrace\bigr|$$denotes the number of nonzero coordinates of $\pmb u$ (often called the ''$\ell_0$ norm'', though it is not a true norm). Equivalently, defining the support$$\text{supp}(\pmb u):=\lbrace i\in[n]:u_i\neq0\rbrace,$$we have that $\pmb{u}$ is $s$-sparse if and only if $|\text{supp}(\pmb u)|\le s$.
>
> >Q2. Extension to non-uniform structures beyond the planted-clique conjecture
>
> **Reply:** We thank the reviewer for positive remark on the non-uniform extension. Since related comments were raised by other reviewers, we expand on this point here and **clarify the motivation and significance of studying the non-uniform setting**.
>
> First, **the planted-clique-based literature only addresses the *uniform* sparse spiked model**. Existing lower bounds based on the planted clique conjecture, such as Brennan et al. (2018), are derived for the canonical homogeneous sparse spiked model, where the support is random and the nonzero coordinates all have equal magnitude (up to signs), so that $||\pmb u||\_\infty=s^{-1/2}$. This symmetry is inherent in the standard planted clique model: conditioned on the planted set, no vertex is distinguished, and every planted edge carries the same excess mean. As a result, these reductions provide hardness evidence for the canonical uniform case, but they do not directly imply hardness for general *non-uniform* sparse spikes with unequal magnitudes. Thus, the planted clique conjecture gives a conjectural computational picture for uniform spikes, but it does not settle the non-uniform setting.
>
> This is exactly why we focus on the non-uniform case. **Our goal is not to resolve the conjectured hardness barrier for the homogeneous model**; under the planted clique conjecture, that statistical-computational gap is already believed to be fundamental. Rather, **we study whether the same barrier persists once the hidden signal departs from perfect uniformity.** Moreover, the non-uniform setting is not made trivial by allowing one coordinate to be much larger: a dominant entry may help identify one location, but it does not by itself yield support recovery or accurate estimation of the full spike. In addition, our result concerns a non-uniform regime quantified by the $\ell_\infty$ condition, rather than relying only on such a degenerate regime.
>
> **Our main contribution is to identify a natural class of non-uniform sparse spikes for which polynomial-time recovery is possible at significantly lower signal strength than in the uniform case.** More specifically, Theorem 2.2 gives a guarantee at signal strength $\widetilde\Omega(\sqrt s||\pmb u||\_\infty^{-1})$ throughout the admissible range $||\pmb u||\_\infty\in[s^{-1/2},1]$. This makes the role of non-uniformity explicit and continuous. In the homogeneous case $||\pmb u||\_\infty=s^{-1/2}$, the requirement becomes $\widetilde\Omega(s)$, which matches the known easy regime and is therefore fully consistent with planted-clique-based hardness. By contrast, when $||\pmb u||\_\infty=\Omega(1)$, our method succeeds at $\widetilde\Omega(\sqrt s)$, matching the information-theoretic scaling known for the homogeneous case. Intermediate regimes interpolate smoothly, e.g. $\widetilde\Omega(s^{1/2+\alpha})$ when $||\pmb u||\_\infty\asymp s^{-\alpha}$ for some $\alpha\in(0,1/2)$.
>
> In this sense, our result should be understood as a *structural separation*: **it does not overcome the planted-clique barrier for planted-clique-derived (uniform) instances, but instead shows that this barrier does not extend to all sparse spikes.** We will revise the manuscript to make this distinction explicit, explain more clearly why planted clique applies only to the uniform case, and better emphasize the role of the $\ell_\infty$ condition and the resulting $\widetilde\Omega(\sqrt s)$ recovery threshold in the favorable non-uniform regime.

---

> > ### Author Rebuttal · Reviewer_ubzF · 2026-04-02
> >
> > I thank the authors for their answer. As mentioned in the original review, I did not raise specific concerns, so I keep the score as is.

---

> > > ### Author Response · Authors · 2026-04-02
> > >
> > > Thank you for your response and for maintaining your score. We appreciate your time and consideration.

---

### Decision · Program_Chairs · 2026-04-30

**Decision:**

Accept (regular)

**Comment:**

The paper studies the sparse spiked Wigner model in the non-uniform sparse regime and proposes a column-thresholding initialization, coupled with truncated power refinement, that improves the required signal strength within this structural class. Reviewers agree that the paper is technically solid, clearly written, and supported by convincing experiments.

The main initial concerns were about significance and positioning, especially the need to clarify why the non-uniform setting is interesting, how the result relates to planted-clique-based hardness for the uniform case, and what is genuinely new relative to prior initialization and refinement methods. The rebuttal addressed these points satisfactorily, and all reviewers indicated that their concerns were resolved.

Overall, the paper makes a meaningful theoretical contribution by identifying a natural regime in which the usual computational barrier does not persist, and by providing a polynomial-time method with rigorous guarantees in that setting. While the scope is more specialized than the canonical uniform case, the final reviewer consensus is clearly positive and supports acceptance.